# Variational Inference with Gaussian Score Matching

**Chirag Modi**
Center for Computational Astrophysics,
Center for Computational Mathematics,
Flatiron Institute, New York
cmodi@flatironinstitute.org

**Charles C. Margossian**
Center for Computational Mathematics,
Flatiron Institute, New York
cmargossian@flatironinstitute.org

**Yuling Yao**
Center for Computational Mathematics,
Flatiron Institute, New York
yyao@flatironinstitute.org

**Robert M. Gower**
Center for Computational Mathematics,
Flatiron Institute, New York
rgower@flatironinstitute.org

**David M. Blei**
Department of Computer Science, Statistics,
Columbia University, New York
david.blei@columbia.edu

**Lawrence K. Saul**
Center for Computational Mathematics,
Flatiron Institute, New York
lsaul@flatironinstitute.org

## Abstract

Variational inference (VI) is a method to approximate the computationally intractable posterior distributions that arise in Bayesian statistics. Typically, VI fits a simple parametric distribution to be close to the target posterior, optimizing an appropriate objective such as the evidence lower bound (ELBO). In this work, we present a new approach to VI. Our method is based on the principle of score matching—namely, that if two distributions are equal then their score functions (i.e., gradients of the log density) are equal at every point on their support. With this principle, we develop score-matching VI, an iterative algorithm that seeks to match the scores between the variational approximation and the exact posterior. At each iteration, score-matching VI solves an inner optimization, one that minimally adjusts the current variational estimate to match the scores at a newly sampled value of the latent variables. We show that when the variational family is a Gaussian, this inner optimization enjoys a closed-form solution, which we call Gaussian score matching VI (GSM-VI). GSM-VI is a "black box" variational algorithm in that it only requires a differentiable joint distribution, and as such it can be applied to a wide class of models. We compare GSM-VI to black box variational inference (BBVI), which has similar requirements but instead optimizes the ELBO. We first study how GSM-VI behaves as a function of the problem dimensionality, the condition number of the target covariance matrix (when the target is Gaussian), and the degree of mismatch between the approximating and exact posterior distribution. We then study GSM-VI on a collection of real-world Bayesian inference problems from the posteriorDB database of datasets and models. We find that GSM-VI is faster than BBVI and equally or more accurate. Specifically, over a wide range of target posteriors, GSM-VI requires 10-100x fewer gradient evaluations than BBVI to obtain a comparable quality of approximation.[1]

---

[1]We provide a Python implementation of GSM-VI algorithm at https://github.com/modichirag/GSM-VI.

# 1  Introduction

This paper is about variational inference for approximate Bayesian computation. Consider a statistical model $p(\boldsymbol{\theta}, \boldsymbol{x})$ of parameters $\boldsymbol{\theta} \in \mathbb{R}^d$ and observations $\boldsymbol{x}$. Bayesian inference aims to infer the posterior distribution $p(\boldsymbol{\theta} \,|\, \boldsymbol{x})$, which is often intractable to compute. Variational inference is an optimization-based approach to approximate the posterior [5, 18].

The idea behind VI is to approximate the posterior with a member of a *variational family* of distributions $q_{\boldsymbol{w}}(\boldsymbol{\theta})$, parameterized by *variational parameters* $\boldsymbol{w}$ [5, 18]. Specifically, VI methods establish a measure of closeness between $q_{\boldsymbol{w}}(\boldsymbol{\theta})$ and the posterior, and then minimize it with an optimization algorithm. Researchers have explored many aspects of VI, including different objectives [8, 9, 20, 26, 27, 29, 34] and optimization strategies [1, 15, 28].

In its modern form, VI typically minimizes $\mathrm{KL}\left(q_{\boldsymbol{w}}(\theta) \| p(\theta \,|\, x)\right)$ with stochastic optimization, and further satisfies the so-called "black-box" criteria [1, 28, 33]. Black-box VI (BBVI) only requires the practitioner to specify the log joint $\log p(\boldsymbol{\theta}, \boldsymbol{x})$ and (often) its gradient $\nabla_{\boldsymbol{\theta}} \log p(\theta, \boldsymbol{x})$, which for many models can be obtained by automatic differentiation [14, 25]. BBVI has been widely implemented, and it is available in many probabilistic programming systems [4, 21, 31].

In this paper, we propose a new approach to VI. We begin with the principle of *score matching* [16], that when two densities are equal then their gradients are equal as well, and we use this principle to derive a new way to fit a variational distribution to be close to the exact posterior. The result is *score-matching VI*. Rather than explicitly minimize a divergence, score-matching VI iteratively projects the variational distribution onto the exact score-matching constraint. This strategy enables a new black-box VI algorithm.

Score-matching VI relies on the same ingredients as reparameterization BBVI [21]—a differentiable variational family and a differentiable log joint—and so it can be as easily incorporated into probabilistic programming systems as well. Further, when the variational family is a Gaussian, score-matching VI is particularly efficient: each iteration is computable in closed form. We call the resulting algorithm Gaussian score matching VI (GSM-VI).

Unlike BBVI, GSM-VI does not rely on stochastic gradient descent (SGD) for its core optimization. Though SGD has the appeal of simplicity, it requires careful tuning of learning rates. GSM-VI was inspired by a different tradition of constraint-based algorithms for online learning [3, 7, 12, 13, 23]. These algorithms have been extensively developed and analyzed for problems in classification, and under the right conditions, they have been observed to outperform SGD. This paper shows how to extend this constraint-based framework—and the powerful machinery behind it—from the problem of classification to the workhorse of Gaussian VI. The key insight is that score-matching (unlike ELBO maximization) lends itself naturally to a constraint-based formulation.

We empirically compared GSM-VI to reparameterization BBVI on several classes of models, and with both synthetic and real-world data. In general, we found that GSM requires 10-100x fewer gradient evaluations to converge to an equally good approximation. When the exact posterior is Gaussian, we found that GSM-VI scales significantly better with respect to dimensionality and is insensitive to the condition number of the target covariance. When the exact posterior is non-Gaussian, we found that GSM-VI converges more quickly without sacrificing the quality of the final approximation.

This paper makes the following contributions:

- We introduce *score-matching variational inference*, a new black-box approach to fitting $q_{\boldsymbol{w}}(\boldsymbol{\theta})$ to be close to $p(\boldsymbol{\theta} \,|\, \boldsymbol{x})$. Score-matching VI requires no tunable optimization hyperparameters, to which BBVI can be sensitive.

- When the variational family is Gaussian, we develop *Gaussian score-matching variational inference* (GSM-VI). It establishes efficient closed-form iterates for score-matching VI.

- We empirically compare GSM-VI to reparameterization BBVI. Across many models and datasets, we found that GSM-VI enjoys faster convergence to an equally good approximation.

We develop score-matching VI in Section 2 and study its performance in Section 3.

**Related work.** This paper introduces a new method for black-box variational inference that relies only on having access to the gradients of the variational distribution and the log joint. GSM-VI has similar goals to automatic-differentiation variational inference (ADVI) [21] and Pathfinder [35], which also fit multivariate Gaussian variational families, but do so by maximizing the ELBO using stochastic optimization. As in GSM-VI, the algorithm of ref. [32] also seeks to match the scores of the variational and the target posterior, but it does so by minimizing the L2 loss between them.

A novel aspect of GSM-VI is how it fits the variational parameters. Rather than minimize a loss function, it aims to solve a set of nonlinear equations. Similar ideas have been pursued in the context of fitting a model to data using empirical risk minimization (ERM). For example, passive agressive (PA) methods [7] and the stochastic polyak stepsize (SPS) are also derived via projections onto sampled nonlinear equations [3, 13, 23]. A probabilistic extension of PA methods is known as confidence-weighted (CW) learning [12]. In CW learning, the learner maintains a multivariate Gaussian distribution over the weight vector of a linear classifier. Similarly, the second step of GSM-VI also minimizes a KL divergence between multivariate Gaussians. Unlike CW learning, however, the projection in GSM-VI enforces a score-matching constraint as opposed to a margin-based constraint for linear classification.

## 2    Score Matching Variational Inference

Suppose for the moment that the variational family $q_{\boldsymbol{w}}(\boldsymbol{\theta})$ is rich enough to perfectly capture the posterior $p(\boldsymbol{\theta} \,|\, \boldsymbol{x})$. In other words, there exists a $\boldsymbol{w}^*$ such that

$$\log q_{\boldsymbol{w}^*}(\boldsymbol{\theta}) = \log p(\boldsymbol{\theta}|\boldsymbol{x}), \qquad \forall \boldsymbol{\theta} \in \Theta. \tag{1}$$

If we could solve Eq. 1 for $\boldsymbol{w}^*$, the resulting variational distribution would be a perfect fit. The challenge is that the posterior on the right side is intractable to compute.

To help, we appeal to the idea of score matching [16]. We define the score of a distribution to be the gradient of its log with respect to the variable[2], e.g., $\nabla_{\boldsymbol{\theta}} \log q_{\boldsymbol{w}}(\boldsymbol{\theta})$. The principle of score matching is that if two distributions are equal at every point in their support then their score functions are also equal.

To use score matching for VI, we first write the log posterior as the log joint minus the normalizing constant, i.e., the marginal distribution of $\boldsymbol{x}$,

$$\log p(\boldsymbol{\theta} \,|\, \boldsymbol{x}) = \log p(\boldsymbol{\theta}, \boldsymbol{x}) - \log p(\boldsymbol{x}). \tag{2}$$

With this expression, the principle of score matching leads to the following Lemma.

> **Lemma 2.1.** Let $\Theta \subset \mathbb{R}^d$ be a set that contains the support of $p(\boldsymbol{\theta}, \boldsymbol{x})$ and $q_{\boldsymbol{w}^*}(\boldsymbol{\theta})$ for some $\boldsymbol{w}^*$. That is $p(\boldsymbol{\theta}, \boldsymbol{x}) = q_{\boldsymbol{w}^*}(\boldsymbol{\theta}) = 0$ for every $\theta \notin \Theta$. Furthermore, suppose that the support of $p(\boldsymbol{\theta}, \boldsymbol{x})$ and $q_{\boldsymbol{w}^*}(\boldsymbol{\theta})$, and $\Theta$ is path-connected. The parameter $\boldsymbol{w}^*$ satisfies
>
> $$\nabla_{\boldsymbol{\theta}} \log q_{\boldsymbol{w}^*}(\boldsymbol{\theta}) = \nabla_{\boldsymbol{\theta}} \log p(\boldsymbol{\theta}, \boldsymbol{x}), \qquad \forall \boldsymbol{\theta} \in \Theta, \tag{3}$$
>
> if and only if $\boldsymbol{w}^*$ also satisfies Eq. 1.

What is notable about Eq. 3 is that the right side is the gradient of the log joint. Unlike the posterior, the gradient of the log joint is tractable to compute for a large class of probabilistic models. (The proof of the lemma is in the appendix.)

This lemma motivates a new algorithm, *score-matching VI*. The idea is to iteratively refine the variational parameters $\boldsymbol{w}$ in order to satisfy (as much as possible) the system of equations in Eq. 3. At each iteration $t$, the algorithm first samples a new $\boldsymbol{\theta}_t$ from the current variational approximation and then minimally adjusts $\boldsymbol{w}$ to satisfy Eq. 3 for that value of $\boldsymbol{\theta}_t$, see (4). We want this adjustment to be minimal because, eventually $q_{\boldsymbol{w}_t}$ will learn a good fit—one that increasingly matches the scores over the support of the target posterior—and we want to preserve this good fit as much as possible.

---

[2]We make this clear because, in some literature, the score is defined as the gradient with respect to the parameter.

> **Score matching variational inference**
>
> At iteration $t$:
>
> 1. Sample $\boldsymbol{\theta}_t \sim q_{\boldsymbol{w}_t}(\boldsymbol{\theta})$.
> 2. Update the variational parameters:
>
> $$\boldsymbol{w}_{t+1} = \arg\min_{\boldsymbol{w}} \mathrm{KL}\left(q_{\boldsymbol{w}_t}(\boldsymbol{\theta}) \,||\, q_{\boldsymbol{w}}(\boldsymbol{\theta})\right)$$
>
> $$\text{such that} \quad \nabla_{\boldsymbol{\theta}} \log q_{\boldsymbol{w}}(\boldsymbol{\theta}_t) = \nabla_{\boldsymbol{\theta}} \log p(\boldsymbol{\theta}_t, \boldsymbol{x}). \tag{4}$$

This algorithm for score matching VI was inspired by earlier online algorithms for learning a classifier from a stream of labeled examples. One particularly elegant algorithm in this setting is known as passive-aggressive (PA) learning [7]; this algorithm incrementally updates its model by the minimal amount to classify each example correctly by a large margin. This approach was subsequently extended to a probabilistic setting, known as confidence-weighted (CW) learning [12] in which one minimally updates a *distribution* over classifiers. Our algorithm is similar in that it minimally updates an approximating distribution for VI, but it is different in that it enforces constraints for score matching instead of large margin classification.

At a high level, what makes this approach to VI likely to succeed or fail? Certainly it is necessary that there are more variational parameters than elements of the latent variable $\boldsymbol{\theta}$; when this is not the case, it may be impossible to satisfy a *single* score matching constraint in Eq. 4. That said, it is standard (as in, e.g., a factorized or mean-field variational family) to have at least as many variational parameters as there are elements of the latent variable. It is also apparent that the algorithm may never converge if the target posterior is not contained in the variational family, or it may converge to a degenerate solution if the variational approximation collapses to a point mass, thus terminating the updates altogether. While we cannot dismiss these possibilities out of hand, we did not observe erratic or pathological outcomes in any of the empirical studies of Section 3.

For more intuition, Figure 1a illustrates the effect of the update in Eq. 4 when both the target and approximating distribution are 1d Gaussian. The target posterior $p(\boldsymbol{\theta} \,|\, \boldsymbol{x})$ is shaded blue. The plot shows the initial variational distribution $q_{\boldsymbol{w}_0}$ (light grey curve) and its update to $q_{\boldsymbol{w}_1}$ (medium grey) so that the gradient of the updated distribution matches the gradient of the target at the sampled $\boldsymbol{\theta}_0$ (dotted red tangent line). It also shows the update from $\boldsymbol{w}_1$ to $\boldsymbol{w}_2$, now matching the gradient at $\boldsymbol{\theta}_1$. With these two updates, $q_{\boldsymbol{w}_2}$ (dark grey) is very close to the target $p(\boldsymbol{\theta} \,|\, \boldsymbol{x})$. With this picture in mind, we now develop the details of this algorithm for a more widely applicable setting.

**Gaussian Score-Matching VI.** Suppose the variational distribution belongs to a multivariate Gaussian family $q_{\boldsymbol{w}}(\boldsymbol{\theta}) := \mathcal{N}(\boldsymbol{\mu}, \boldsymbol{\Sigma})$; this is a common choice especially in systems for automated approximate inference [1, 21]. One of our main contributions is to show in this case that Eq. 4 has a closed-form solution. The solution $\boldsymbol{w}_{t+1} = (\boldsymbol{\mu}_{t+1}, \boldsymbol{\Sigma}_{t+1})$ has the following form:

$$\boldsymbol{\mu}_{t+1} = \boldsymbol{\mu}_t + \boldsymbol{A}_t \left(\nabla_{\boldsymbol{\theta}} \log p(\boldsymbol{\theta}_t, \boldsymbol{x}) - \nabla_{\boldsymbol{\theta}} \log q_{\boldsymbol{w}_t}(\boldsymbol{\theta}_t)\right) \tag{5}$$

$$\boldsymbol{\Sigma}_{t+1} = \boldsymbol{\Sigma}_t + (\boldsymbol{\mu}_t - \boldsymbol{\theta}_t)(\boldsymbol{\mu}_t - \boldsymbol{\theta}_t)^\top - (\boldsymbol{\mu}_{t+1} - \boldsymbol{\theta}_t)(\boldsymbol{\mu}_{t+1} - \boldsymbol{\theta}_t)^\top \tag{6}$$

where $\boldsymbol{A}_t \in \mathbb{R}^{d \times d}$ is a matrix defined in the theorem below. Note that these updates, which exactly solve the optimization in Eq. 4, only require the score of the log joint $\nabla_{\boldsymbol{\theta}} \log p(\boldsymbol{\theta}, \boldsymbol{x})$ and the score of the variational distribution $\nabla_{\boldsymbol{\theta}} \log q_{\boldsymbol{w}}(\boldsymbol{\theta})$.

Before deriving Eqs. 5 and 6 in full, we highlight the intuitive form of this closed-form solution. Consider the approximation at the $t$th iteration $q_{\boldsymbol{w}_t}$ and the current sample $\boldsymbol{\theta}_t$. First suppose the scores already match at this sample, that is $\nabla_{\boldsymbol{\theta}} \log p(\boldsymbol{\theta}_t, \boldsymbol{x}) = \nabla_{\boldsymbol{\theta}} \log q_{\boldsymbol{w}_t}(\boldsymbol{\theta}_t)$. Then the mean does not change $\boldsymbol{\mu}_{t+1} = \boldsymbol{\mu}_t$ and, similarly, the two rank-one terms in the covariance update in Eq. 6 cancel out so $\boldsymbol{\Sigma}_{t+1} = \boldsymbol{\Sigma}_t$. This shows that when $q_{\boldsymbol{w}_t}(\boldsymbol{\theta}) = p(\boldsymbol{\theta}, \boldsymbol{x})$ for all $\boldsymbol{\theta}$, the method stops. On the other hand, if the scores do not match, then the mean is updated proportionally to the difference between the scores, and the covariance is updated by a rank-two correction. Figure 1b illustrates the vector field of updates for a one dimensional target $p(\boldsymbol{\theta}, \boldsymbol{x}) = \mathcal{N}(0, 1)$. The vector field points to the solution (green star), which is also the only fixed point of the updates.

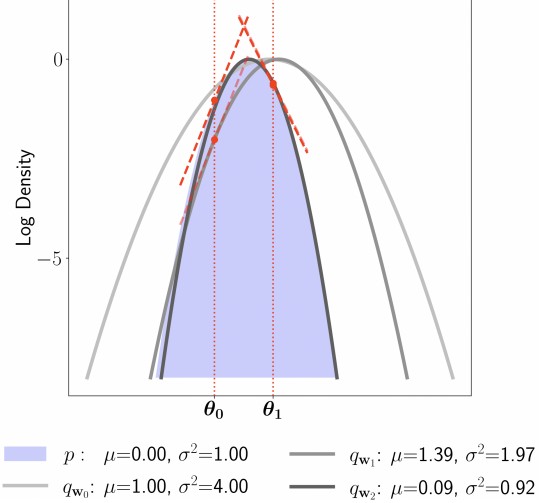

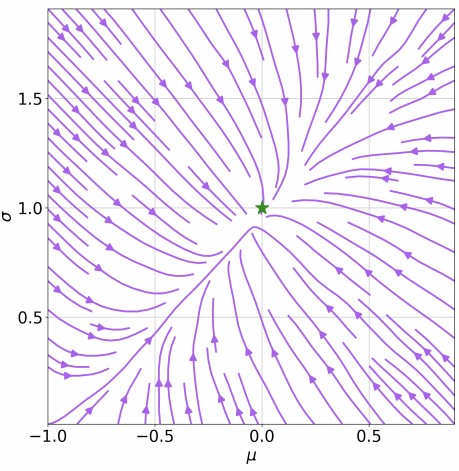

(a) Two iterations of GSM-VI. The log density of the target posterior $p$ is shaded blue; the initial distribution $q_{\boldsymbol{w}_0}$ is light grey; the first update $q_{\boldsymbol{w}_1}$ is medium grey; and the second update $q_{\boldsymbol{w}_2}$ is dark grey.

(b) The vector field of Eq. 4, averaged over 5 independent samples, where $p(\boldsymbol{\theta} \mid \boldsymbol{x}) = \mathcal{N}(0, 1)$. The solution $(\mu, \sigma) = (0, 1)$ is the green star.

We now formalize this result and give the complete expression for the matrix $\boldsymbol{A}_t$ that appears in the update for the mean of the variational approximation in Eq. 5.

**Theorem 2.2. (GSM-VI updates)** Let $p(\boldsymbol{\theta}, \boldsymbol{x})$ be given for some $\boldsymbol{\theta} \in \mathbb{R}^d$, and let $q_{\boldsymbol{w}^t}(\boldsymbol{\theta})$ and $q_{\boldsymbol{w}}(\boldsymbol{\theta})$ be multivariate normal distributions with means $\boldsymbol{\mu}_t$ and $\boldsymbol{\mu}$ and covariance matrices $\boldsymbol{\Sigma}_t$ and $\boldsymbol{\Sigma}$, respectively. As shorthand, let $\boldsymbol{g}_t := \nabla_{\boldsymbol{\theta}} \log p(\boldsymbol{\theta}_t, \boldsymbol{x})$ and let

$$\boldsymbol{\mu}_{t+1}, \boldsymbol{\Sigma}_{t+1} = \underset{\boldsymbol{\mu}, \boldsymbol{\Sigma} \succeq 0}{\operatorname{argmin}} \left[ \mathrm{KL}(q_t, q) \right] \quad \text{such that} \quad \nabla_{\boldsymbol{\theta}} \log q(\boldsymbol{\theta}_t) = \nabla_{\boldsymbol{\theta}} \log p(\boldsymbol{\theta}_t, \boldsymbol{x}). \quad (7)$$

The solution to eq. (7) is given by Eqs. 5 and 6 where

$$\boldsymbol{A}_t := \frac{1}{1+\rho} \left[ \mathbf{I} - \frac{(\boldsymbol{\mu}_t - \boldsymbol{\theta}_t)\boldsymbol{g}_t^{\top}}{1 + \rho + (\boldsymbol{\mu}_t - \boldsymbol{\theta}_t)^{\top}\boldsymbol{g}_t} \right] \boldsymbol{\Sigma}_t, \quad (8)$$

and $\rho$ is the positive root of the quadratic equation

$$\rho(1+\rho) = \boldsymbol{g}_t^{\top}\boldsymbol{\Sigma}_t\boldsymbol{g}_t + \left[ (\boldsymbol{\mu}_t - \boldsymbol{\theta}_t)^{\top}\boldsymbol{g}_t \right]^2. \quad (9)$$

With the definition of $\boldsymbol{A}_t$ in Eq. 8 we can see that the computational complexity of updating $\boldsymbol{\mu}$ and $\boldsymbol{\Sigma}$ via Eqs. 5 and 6 is $\mathcal{O}(d^2)$, where $\boldsymbol{\theta} \in \mathbb{R}^d$ and we assume the cost of computing the gradients is $\mathcal{O}(d)$. Note this is the best possible iteration complexity we can hope for, since we store and maintain the full covariance matrix of $d^2$ elements. (The proof is in the appendix.)

Algorithm 1 presents the full GSM-VI algorithm. Here we also use mini-batching, where we average over $B \in \mathbb{N}$ independently sampled updates of Eqs. 5 and 6 before updating the mean and covariance.

## 3 Empirical Studies

We evaluate how well GSM-VI can approximate a variety of target posterior distributions. Recall that GSM-VI uses a multivariate Gaussian distribution as its variational family. We investigate two separate cases—one when the target posterior is in this family, and the other when it is not.

**Algorithmic details and comparisons.** We compare GSM-VI with a reparameterization variant of BBVI as the baseline, similar to [21]. BBVI uses the same multivariate Gaussian variational family, which we fit by maximizing the ELBO. (Maximizing the ELBO is equivalent to minimizing KL).

**Algorithm 1:** Gaussian Score Matching VI

---

**Input** : Initial mean estimate $\boldsymbol{\mu}_0$, initial covariance estimate $\boldsymbol{\Sigma}_0$, target distribution $p(\boldsymbol{\theta}|\boldsymbol{x})$,
number of iterations $N \in \mathbb{N}$, batch size $B \in \mathbb{N}$.
**Output** : Multivariate normal variational distribution $q_{\boldsymbol{w}}(\boldsymbol{\theta}) := \mathcal{N}(\boldsymbol{\mu}, \boldsymbol{\Sigma})$
**for** $i = 0, \ldots, N-1$   ▷ iteration loop
**do**
    **for** $j = 0, \ldots, B-1$   ▷ batch loop
    **do**
        Sample $\boldsymbol{\theta}^{(j)} \sim \mathcal{N}(\boldsymbol{\mu_i}, \boldsymbol{\Sigma_i})$
        $\boldsymbol{g} \leftarrow \nabla_{\boldsymbol{\theta}} \log p(\boldsymbol{\theta}^{(j)}|\boldsymbol{x})$
        $\boldsymbol{\varepsilon} \leftarrow \boldsymbol{\Sigma}_i \boldsymbol{g} - \boldsymbol{\mu}_i + \boldsymbol{\theta}$
        Solve $\rho(1+\rho) = \boldsymbol{g}^{\top}\boldsymbol{\Sigma}_i\boldsymbol{g} + \left[(\boldsymbol{\mu}_i - \boldsymbol{\theta})^{\top}\boldsymbol{g}\right]^2$ for $\rho > 0$
        $\boldsymbol{\delta\mu}^{(j)} \leftarrow \frac{1}{1+\rho}\left[\mathbf{I} - \frac{(\boldsymbol{\mu}_i - \boldsymbol{\theta})\boldsymbol{g}^{\top}}{1+\rho+(\boldsymbol{\mu}_i - \boldsymbol{\theta})^{\top}\boldsymbol{g}}\right]\boldsymbol{\varepsilon}$
        $\boldsymbol{\mu}_i^{(j)} \leftarrow \boldsymbol{\mu}_i + \boldsymbol{\delta\mu}^{(j)}$
        $\boldsymbol{\delta\Sigma}^{(j)} \leftarrow (\boldsymbol{\mu}_i - \boldsymbol{\theta})(\boldsymbol{\mu}_i - \boldsymbol{\theta})^{\top} - (\boldsymbol{\mu}_i^{(j)} - \boldsymbol{\theta})(\boldsymbol{\mu}_i^{(j)} - \boldsymbol{\theta})^{\top}$
    **end**
    Update $\boldsymbol{\mu_{i+1}} \leftarrow \boldsymbol{\mu}_i + \sum_j \boldsymbol{\delta\mu^{(j)}}/B$
    Update $\boldsymbol{\Sigma_{i+1}} \leftarrow \boldsymbol{\Sigma}_i + \sum_j \boldsymbol{\delta\Sigma^{(j)}}/B$
**end**
$q_{\boldsymbol{w}}(\boldsymbol{\theta}) \leftarrow \mathcal{N}(\boldsymbol{\mu}_N, \boldsymbol{\Sigma}_N)$

---

**Algorithm 2:** Black-box variational inference

---

**Input** : Initial mean estimate $\boldsymbol{\mu_0}$, Initial covariance estimate $\boldsymbol{\Sigma_0}$, target distribution $p(\boldsymbol{\theta}|\boldsymbol{x})$,
number of iterations N, batch size B, learning rate $\epsilon$
**Output** : Multivariate normal variational distribution $q_{\boldsymbol{w}}(\boldsymbol{\theta}) := \mathcal{N}(\boldsymbol{\mu}, \boldsymbol{\Sigma})$
$q_{\boldsymbol{w}} \leftarrow \mathcal{N}(\boldsymbol{\mu_0}, \boldsymbol{\Sigma_0})$ ;
**for** $i = 0, \ldots, N-1$   ▷ iteration loop
**do**
    $\{\boldsymbol{\theta}^{(0)}, \boldsymbol{\theta}^{(1)}, ..., \boldsymbol{\theta}^{(B)}\} \sim q_{\boldsymbol{w}}(\boldsymbol{\theta})$   ▷ Sample a batch of B points;
    $\text{ELBO} = \sum_j \log(p(\boldsymbol{\theta}^{(j)}, \boldsymbol{x}) - \log q_{\boldsymbol{w}}(\theta^{(j)})$ ;
    $\boldsymbol{w} \leftarrow \boldsymbol{w} - \epsilon\nabla_{\boldsymbol{w}}\text{ELBO}$   ▷ Optimization step, we use ADAM;
**end**

---

We use the ADAM optimizer [19] with default settings but vary the learning rate between $10^{-1}$ and $10^{-3}$. We report results only for the best performing setting. The full BBVI algorithm is shown in Algorithm 2.

The only free parameter in GSM-VI is the batch size $B$. We find that for Gaussian targets, GSM performs equally well for all $B \geq 1$. There are marginal gains for larger batches for high dimensional targets, but $B = 2$ is a good conservative default. For non-Gaussian targets, we find that all batch sizes converge to the same solution, but we also note that larger batch sizes help to dampen the oscillations in KL divergence around the optimal solution. Notwithstanding these effects, it is not generally necessary to fine-tune this hyperparameter. In all studies of this section we report results for $B = 2$ and show that it is a good default baseline. We use the same batch size for BBVI.

Both GSM-VI and BBVI require an initial distribution for the variational approximation. Unless specified otherwise, we initialize the variational approximation as a Gaussian distribution with zero mean and identity covariance matrix.

**Evaluation metric.** GSM-VI does not explicitly minimize any loss function. Hence to compare its performance against BBVI, we estimate empirical divergences between the variational and the target distribution and show their evolution with the number of gradient evaluations. We measure performance in terms of gradient evaluations as the actual running time can be sensitive

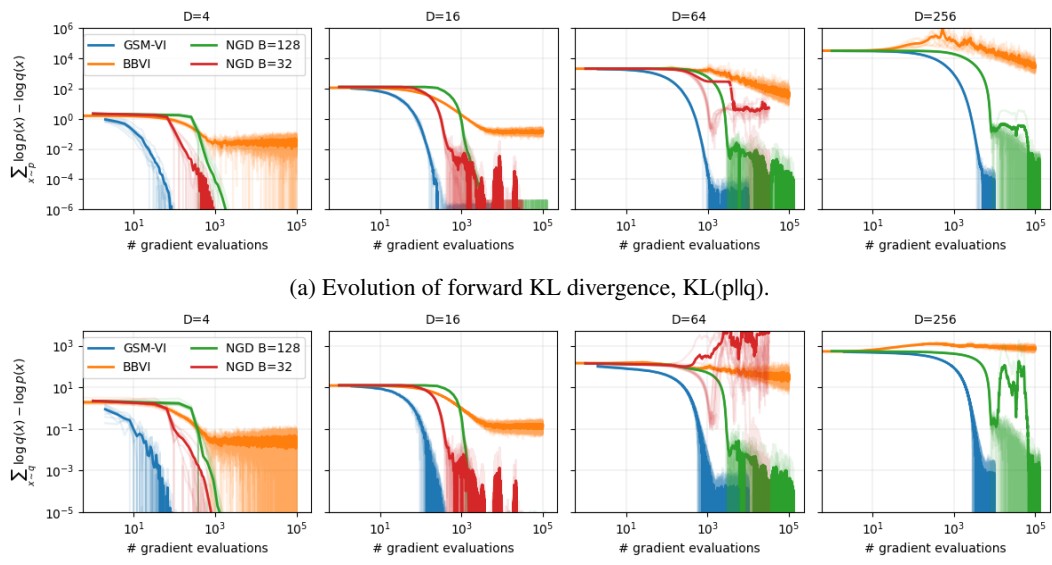

(a) Evolution of forward KL divergence, KL(p‖q).

(b) Evolution of reverse KL divergence, KL(q‖p).

Figure 2: Scaling with dimensionality: evolution of forward (a) and reverse KL (b) with the number of gradient evaluations of the target distribution, which here is a Gaussian distribution with a dense $D \times D$ covariance matrix. Each panel shows the results for a different dimensionality $D$ (indicated in the title). Translucent lines show the scatter of 10 different runs, and the solid line shows the average. GSM-VI and BBVI use batch sizes of $B = 2$, while the batch sizes for natural gradient descent are shown in the legend.

to implementation details; moreover, we note that in real-world problems, the computation is often bottlenecked by evaluations of the target log density and its gradient. For completeness, we also measure running times of GSM-VI and BBVI for a full rank Gaussian target with 2048 dimensions using sample size of 1 in Jax after JIT-compilation. The GSM update takes 230 ms/update while BBVI takes 227 ms/update.

In the experiments with synthetic models in Sections 3.1, and 3.2 we have access to the true distribution; hence for these experiments, we estimate the forward KL divergence (FKL) empirically by $\sum_{\boldsymbol{\theta}_i \sim p(\boldsymbol{\theta})} \log p(\boldsymbol{\theta}_i) - \log q(\boldsymbol{\theta}_i)$. To reduce stochasticity, we always use the same pre-generated set of 1000 samples from the target distribution. In the experiments with real-world models in Section 3.3, we do not have access to the samples from the target distribution; for these experiments we monitor the negative ELBO. In all experiments, we show the results for 10 independent runs.

## 3.1 GSM-VI for Gaussian approximation

We begin by studying GSM-VI where the target distribution is also a multivariate Gaussian.

**Scaling with dimensions.** How does GSM-VI scale with respect to the dimensionality, $D$, of the latent variable? Figure 2 shows the convergence of FKL for GSM-VI and BBVI for different dimensionalities. Empirically, for GSM-VI, we find that the number of iterations required for convergence increases almost linearly with dimensionality. The scaling for BBVI is worse: even for small problems ($D < 64$) it requires 100 times more iterations while also converging to a sub-optimal solution (as measured by the FKL metric). The figure also shows the evolution of the *reverse* KL divergence, and it is noteworthy that GSM-VI also outperforms BBVI in this metric.

For completeness, we also experimented on these problems[3] with methods for variational inference based on natural gradient descent (NGD-VI) [2, 22]. Natural gradient descent is a stochastic gradient method that uses the Fisher matrix as a preconditioner. We find NGD-VI is slower by factor of

---

[3]Specifically, we use the package GMMVI with default settings except changing the batch size.

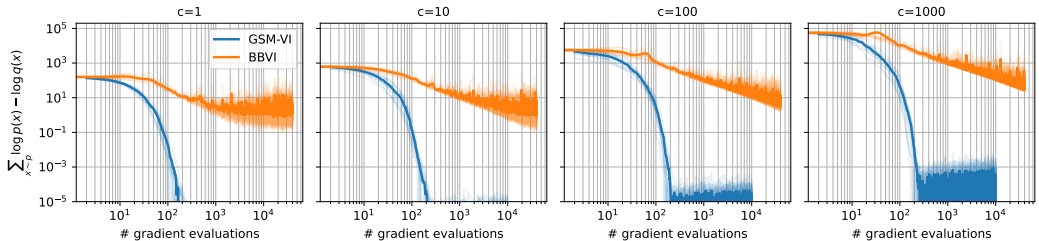

Figure 3: Impact of condition number: evolution of FKL with the number of gradient evaluations of the target distribution. The target is a 10-dimensional Gaussian albeit with a dense covariance matrix of different condition numbers $c$ specified in the title of different panels. Translucent lines show the scatter of 10 different runs and the solid line shows the average.

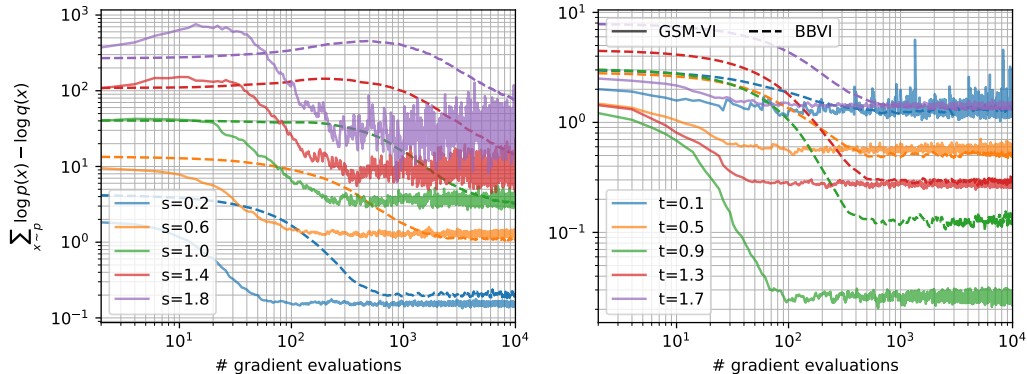

Figure 4: Impact of non-Gaussianity: evolution of FKL with the number of gradient evaluations for Sinh-arcsinh distributions with 10-dimensional dense Gaussian as the base distribution. Gaussian distribution has $s = 0$, $t = 1$. In the left panel, we vary skewness $s$ while fixing $t = 1$, and in the right panel we vary the tail-weight $t$ with skewness fixed to $s = 0$. Solid lines are the results for GSM, dashed for BBVI.

5-10x than GSM-VI. While NGD-VI outperforms BBVI, its performance is sensitive to the choice of batch size, with small batch sizes prone to divergence. Furthermore, we note that that per-iteration-complexity of NGD-VI is cubic in the dimensionality ($D^3$) as each iteration requires inverting a Hessian approximation; this is in contrast to the quadratic ($D^2$) per-iteration-complexities of GSM-VI and BBVI. For these reasons, we do not investigate NGD-VI further in this paper.

**Impact of condition number.** What is the impact of the shape of the target distribution? We again consider a Gaussian target distribution, but vary the condition number of its covariance matrix by fixing its smallest eigenvalue to $0.1$ and scaling the largest eigenvalue to $0.1 \times c$. Figure 3 shows the results for a 10 dimensional Gaussian where we vary the condition number $c$ from 1 to 1000. Convergence of GSM-VI seems to be largely insensitive to the condition number of the covariance matrix. BBVI on the other hand struggles with poorly conditioned problems, and it does not converge for $c > 100$ even with 100 times more iterations than GSM.

## 3.2  GSM-VI for non-Gaussian target distributions

GSM-VI was designed to solve the exact score-matching equations in Eq. 3, but these equations only have a solution when the family of variational distributions contains the target distribution (see Lemma 2.1). Here we investigate the sensitivity of GSM-VI to this assumption by fitting non-Gaussian target distributions with varying degrees on non-Gaussianity. Specifically, we suppose that

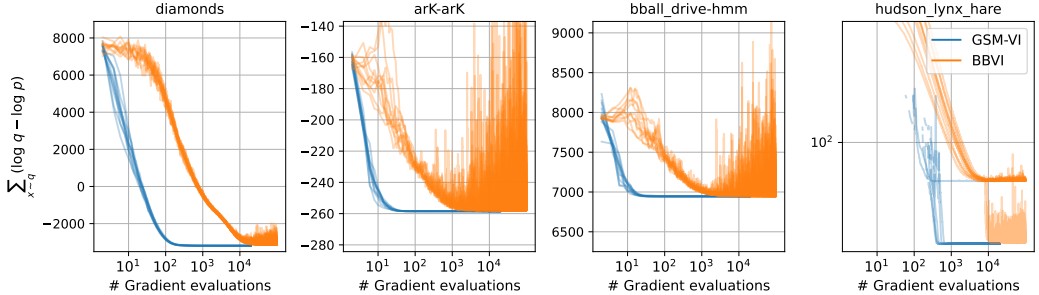

Figure 5: Models from `posteriordb`: Convergence of the ELBO for four models with multivariate normal posteriors. We show results for 10 runs.

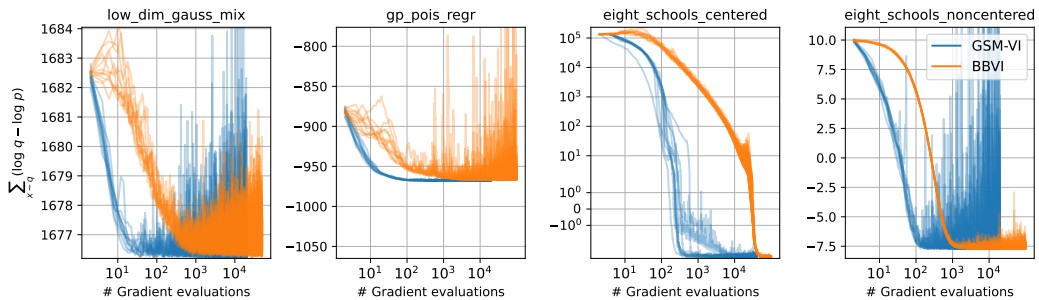

Figure 6: Models from `posteriordb`: Covergence of ELBO for four models with non-Gaussian posteriors. We show results for 10 runs.

the target has a multivariate `Sinh-arcsinh` normal distribution [17]

$$\boldsymbol{z} \sim \mathcal{N}(\boldsymbol{\mu}, \boldsymbol{\Sigma}); \quad \boldsymbol{x} = \sinh\left(\frac{1}{t}\left[\sinh^{-1}(\boldsymbol{z}) + s\right]\right) \tag{10}$$

where the scalar parameters $s$ and $t$ control, respectively, the skewness and the heaviness of the tails. Note that Eq. 10 reduces as a special case to a Gaussian distribution for the choices $s = 0$ and $t = 1$ of these parameters.

Figure 4 shows the result for fitting the variational Gaussian to a 10-dimensional Sinh-arcsinh normal distribution for different values of $s$ and $t$. As the target departs further from Gaussianity, the quality of variational fit worsens for both GSM-VI (solid lines) and BBVI (dashed lines), but they converge to a fit of similar quality in terms of average FKL. GSM converges to this solution at least 10 times faster than BBVI. For highly non-Gaussian targets ($s \geq 1$ or $|t - 1| \geq 0.8$), we have found that GSM-VI does not converge to a fixed point, and it can experience oscillations that are larger in amplitude than BBVI; see for instance $s = 1.8$ and $t = 0.1$ on the left and right of Figure 4, respectively.

### 3.3 GSM-VI on real-world data

We evaluate GSM-VI for approximate on real-world data with 8 models from the `posteriordb` database [24]. The database provides the `Stan` code, data and reference posterior samples, and we use `bridgestan` to access the gradients of these models [6, 30]. We study the following statistical models, each with a different complexity and dimensionality (D) : `diamonds` (generalized linear models, D=26), `hudson-lynx-hare` (differential equation dynamics, D=8), `bball-drive` (hidden Markov models, D=8) and `arK` (time-series, D=7), `eight-schools-centered` and `non-centered` (hierarchical meta-analysis, D=10), `gp-pois-regr` (Gaussian processes, D=13), `low-dim-gauss-mix` (Gaussian mixture, D=5).

For each model (except `hudson-lynx-hare`), we initialize the variational parameter $\boldsymbol{\mu}_0$ at the mode of the distribution, and we set $\boldsymbol{\Sigma}_0 = 0.1\,\boldsymbol{I}_d$ where $\boldsymbol{I}_d$ is the identity matrix of dimension $d$.

For `hudson-lynx-hare`, we initialize the variational distribution as standard normal. We also experimented with other initializations. We find that they do not qualitatively change the conclusions, but can have larger variance between different runs.

We show the evolution of the ELBO for 10 runs of these models. Four of the models have posteriors that can be fit with multivariate normal distribution: `diamonds`, `hudson-lynx-hare`, `bball-drive`, and `arK`. Figure 5 shows the result for these models. The other models have non-Gaussian posteriors: `eight-schools-centered`, `eight-schools-non-centered`, `gp-pois-regr`,, and `low-dim-gauss-mix`. Figure 6 shows the results.

Overall, GSM-VI outperforms BBVI by a factor of 10-100x. When the target posterior is Gaussian, GSM-VI leads to more stable solutions. When the target is non-Gaussian, it converges to the same quality of variational approximation as BBVI. Finally, we note that while GSM-VI yields noisy estimates of the ELBO, it does yield stable estimates of the 1-D marginals and their moments.

## 4    Conclusion and Future Work

In this paper we have proposed Gaussian score matching VI (GSM-VI), a new approach for VI when the variational family is multivariate Gaussian. GSM-VI does not explicitly minimize a divergence between the variational and target distribution; instead, it repeatedly enforces score-matching constraints with closed-form updates for the mean and covariance matrix of the variational distribution. This is in contrast to stochastic gradient based methods, such as in BBVI, that rely on first-order Taylor approximations of their objective function.

GSM-VI is implemented in an open-source Python code at https://github.com/modichirag/GSM-VI.

Unlike approaches that are rooted in stochastic gradient descent, GSM-VI does not require the tuning of step-size hyper-parameters. Rather, it is able to adaptively make large jumps in the initial iterations (see Fig. 1a) and make smaller adjustments as the approximation converges to the target. GSM-VI has only one free parameter, the batch size, and we found a batch-size of 2 to perform competitively across all experiments. Another choice is how to initialize the variational distribution. For the experiments in this paper, we initialized the covariance matrix as the identity matrix, but additional gains could potentially be made with more informed choices derived from a Laplace approximation or L-BFGS Hessian approximation [35].

We evaluated the performance of GSM-VI on synthetic targets and real-world models from `posteriordb`. In general, we found that it requires 10-100x fewer gradient evaluations than BBVI for the target distribution to converge. When the target distribution is itself multivariate Gaussian, we observed that GSM-VI scales nearly *linearly* with dimensionality, which is significantly better than BBVI, and that GSM-VI is fairly insensitive to the condition number of the target covariance matrix. We also found that GSM-VI converges more quickly than BBVI—and also to a solution with a larger ELBO, which is surprising given that BBVI explicitly maximizes the ELBO.

An important avenue for future work is to provide a proof that GSM-VI converges. We note that good convergence results have been obtained for analogous methods that project onto interpolation equations for empirical risk minimization. For instance the Stochastic Polyak Step achieves the min-max optimal rates of convergence for SGD [23]. Note that convergence of VI is a generally challenging problem, with no known rates of convergence even for BBVI [10, 11]. This and others directions are left to future work.

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

## A Proof of Lemma 2.1

**Lemma 2.1.** Let $\Theta \subset \mathbb{R}^d$ be a set that contains the support of $p(\boldsymbol{\theta}, \boldsymbol{x})$ and $q_{\boldsymbol{w}^*}(\boldsymbol{\theta})$ for some $\boldsymbol{w}^*$. That is $p(\boldsymbol{\theta}, \boldsymbol{x}) = q_{\boldsymbol{w}^*}(\boldsymbol{\theta}) = 0$ for every $\theta \notin \Theta$. Furthermore, suppose that the support of $p(\boldsymbol{\theta}, \boldsymbol{x})$ and $q_{\boldsymbol{w}^*}(\boldsymbol{\theta})$, and $\Theta$ is path-connected. The parameter $\boldsymbol{w}^*$ satisfies

$$\nabla_{\boldsymbol{\theta}} \log q_{\boldsymbol{w}^*}(\boldsymbol{\theta}) = \nabla_{\boldsymbol{\theta}} \log p(\boldsymbol{\theta}, \boldsymbol{x}), \qquad \forall \boldsymbol{\theta} \in \Theta, \tag{3}$$

if and only if $\boldsymbol{w}^*$ also satisfies Eq. 1.

*Proof.* (1) $\implies$ (3): Differentiating both sides of (1) in $\boldsymbol{\theta}$ gives

$$\nabla_\theta \log q_{\boldsymbol{w}^*}(\boldsymbol{\theta}) = \nabla_\theta \log p(\boldsymbol{\theta}|\boldsymbol{x}) = \nabla_\theta \left( \log p(\boldsymbol{\theta}, \boldsymbol{x}) - \log p(\boldsymbol{x}) \right)$$
$$= \nabla_\theta \log p(\boldsymbol{\theta}, \boldsymbol{x}), \qquad \forall \boldsymbol{\theta} \in \Theta.$$

(3) $\implies$ (1): Let $\boldsymbol{\theta}_0$ be any arbitrary point in $\Theta$. Because $\Theta$ is path-connected, every $\boldsymbol{\theta} \in \Theta$ is connected to $\boldsymbol{\theta}_0$ via a differentiable path $\mathbf{r}(t)$ where $\mathbf{r}(0) = \boldsymbol{\theta}_0$ and $\mathbf{r}(1) = \boldsymbol{\theta}$. Integrating both sides of (3) along this path $\mathbf{r}(t)$, using again that $\nabla_\theta \log p(\boldsymbol{\theta}|\boldsymbol{x}) = \nabla_\theta \log p(\boldsymbol{\theta}, \boldsymbol{x})$, and using the Fundamental Theorem of Calculus gives

$$\log q_{\boldsymbol{w}^*}(\boldsymbol{\theta}) - \log q_{\boldsymbol{w}^*}(\boldsymbol{\theta}_0) = \int_0^1 \langle \nabla_\theta \log q_{\boldsymbol{w}^*}(\mathbf{r}(t)), \mathbf{r}'(t) \rangle \, dt$$

$$= \int_0^1 \langle \nabla_\theta \log p((\mathbf{r}(t))|\boldsymbol{x}), \mathbf{r}'(t) \rangle \, dt$$

$$= \log p(\boldsymbol{\theta}|\boldsymbol{x}) - \log p(\boldsymbol{\theta}_0|\boldsymbol{x}), \quad \forall \boldsymbol{\theta} \in \Theta.$$

Rearranging and defining $C := \log q_{\boldsymbol{w}^*}(\boldsymbol{\theta}_0) - \log p(\boldsymbol{\theta}_0|\boldsymbol{x})$ gives

$$\log q_{\boldsymbol{w}^*}(\boldsymbol{\theta}) = \log p(\boldsymbol{\theta}|\boldsymbol{x}) + C, \qquad \forall \boldsymbol{\theta} \in \Theta,$$

By exponentiating both sides and integrating in $\boldsymbol{\theta}$ over $\Theta$ we have that

$$1 = \int_{\boldsymbol{\theta}\in\Theta} q_{\boldsymbol{w}^*}(\boldsymbol{\theta})d\boldsymbol{\theta} = e^C \int_{\boldsymbol{\theta}\in\Theta} p(\boldsymbol{\theta}|\boldsymbol{x})d\boldsymbol{\theta} = e^C,$$

where we used that $\Theta$ contains the support of both $p(\boldsymbol{\theta}|\boldsymbol{x})$ and $q_{\boldsymbol{w}^*}(\boldsymbol{\theta})$. Consequently $C = 0$, which gives our result. $\qquad\square$

## B  Proof of Theorem 2.2

Here we give the proof for Theorem 2.2. We also re-introduce the theorem with a simplified notation, where we use $(\boldsymbol{\mu}_0, \boldsymbol{\Sigma}_0)$ to denote the mean and covariance at the previous time step of the method, thus dropping the iteration counter $t$.

**Theorem B.1. (GSM updates)** Let $p(\boldsymbol{\theta}, \boldsymbol{x})$ be given for some $\boldsymbol{\theta} \in \mathbb{R}^d$, and let $q_0(\boldsymbol{\theta})$ and $q(\boldsymbol{\theta})$ be the multivariate normal distributions, respectively, with means $\boldsymbol{\mu}_0$ and $\boldsymbol{\mu}$ and covariance matrices $\boldsymbol{\Sigma}_0$ and $\boldsymbol{\Sigma}$. We seek the distribution

$$\arg\min_{\boldsymbol{\mu},\boldsymbol{\Sigma}=\boldsymbol{\Sigma}^\top} \left[\mathrm{KL}(q_0, q)\right] \quad \text{such that} \quad \nabla_{\boldsymbol{\theta}} \log q(\boldsymbol{\theta}) = \nabla_{\boldsymbol{\theta}} \log p(\boldsymbol{\theta}, \boldsymbol{x}). \tag{11}$$

As shorthand, let $\boldsymbol{g} := \nabla_{\boldsymbol{\theta}} \log p(\boldsymbol{\theta}, \boldsymbol{x})$, and let $\rho$ be the positive root of the quadratic equation

$$\rho(1+\rho) = \boldsymbol{g}^\top \boldsymbol{\Sigma}_0 \boldsymbol{g} + \left[(\boldsymbol{\mu}_0 - \boldsymbol{\theta})^\top \boldsymbol{g}\right]^2. \tag{12}$$

Then the solution is given by the following closed-form updates:

$$\boldsymbol{\mu} = \boldsymbol{\mu}_0 + \frac{1}{1+\rho} \left[\mathbf{I} - \frac{(\boldsymbol{\mu}_0 - \boldsymbol{\theta})\boldsymbol{g}^\top}{1 + \rho + (\boldsymbol{\mu}_0 - \boldsymbol{\theta})^\top \boldsymbol{g}}\right] \boldsymbol{\Sigma}_0 \left(\boldsymbol{g} - \nabla_{\boldsymbol{\theta}} \log q_0(\boldsymbol{\theta})\right), \tag{13}$$

$$\boldsymbol{\Sigma} = \boldsymbol{\Sigma}_0 + (\boldsymbol{\mu}_0 - \boldsymbol{\theta})(\boldsymbol{\mu}_0 - \boldsymbol{\theta})^\top - (\boldsymbol{\mu} - \boldsymbol{\theta})(\boldsymbol{\mu} - \boldsymbol{\theta})^\top. \tag{14}$$

Furthermore, if $\boldsymbol{\Sigma}_0$ is symmetric positive definite then so is $\boldsymbol{\Sigma}$.

*Proof.* The constraint in this optimization is given by

$$\boldsymbol{g} = \nabla_{\boldsymbol{\theta}} \log q(\boldsymbol{\theta}) \tag{15}$$

$$= \nabla_{\boldsymbol{\theta}} \left[-\tfrac{1}{2}(\boldsymbol{\theta} - \boldsymbol{\mu})\boldsymbol{\Sigma}^{-1}(\boldsymbol{\theta} - \boldsymbol{\mu}) - \tfrac{1}{2}\log\left((2\pi)^d|\boldsymbol{\Sigma}|\right)\right] \tag{16}$$

$$= -\boldsymbol{\Sigma}^{-1}(\boldsymbol{\theta} - \boldsymbol{\mu}). \tag{17}$$

The KL divergence is given by

$$\mathrm{KL}(q_0, q) = \frac{1}{2}\left\{\mathrm{tr}[\boldsymbol{\Sigma}^{-1}\boldsymbol{\Sigma}_0] + \log\frac{|\boldsymbol{\Sigma}|}{|\boldsymbol{\Sigma}_0|} + (\boldsymbol{\mu} - \boldsymbol{\mu}_0)^\top \boldsymbol{\Sigma}^{-1}(\boldsymbol{\mu} - \boldsymbol{\mu}_0) - d\right\}. \tag{18}$$

Dropping irrelevant terms from the optimization, we obtain the Lagrangian

$$\mathcal{L}(\boldsymbol{\mu}, \boldsymbol{\Sigma}, \boldsymbol{\lambda}) = \frac{1}{2}\left\{\mathrm{tr}[\boldsymbol{\Sigma}^{-1}\boldsymbol{\Sigma}_0] - \log|\boldsymbol{\Sigma}^{-1}| + (\boldsymbol{\mu} - \boldsymbol{\mu}_0)^\top \boldsymbol{\Sigma}^{-1}(\boldsymbol{\mu} - \boldsymbol{\mu}_0)\right\} + \boldsymbol{\lambda}^\top\left(\boldsymbol{g} - \boldsymbol{\Sigma}^{-1}(\boldsymbol{\mu} - \boldsymbol{\theta})\right). \tag{19}$$

It is easier to optimize the matrix $\boldsymbol{\Sigma}^{-1}$ instead of $\boldsymbol{\Sigma}$. We can enforce the symmetry of $\boldsymbol{\Sigma}^{-1}$ by writing

$$\boldsymbol{\Sigma}^{-1} = \tfrac{1}{2}\left(\boldsymbol{\Phi} + \boldsymbol{\Phi}^\top\right) \tag{20}$$

and performing an unconstrained optimization over $\boldsymbol{\Phi}$. With respect to the latter, the gradients of the Lagrangian are given by

$$\frac{\partial\mathcal{L}}{\partial\Phi_{ij}} = \sum_{kl}\left(\frac{\partial\mathcal{L}}{\partial\Sigma_{kl}^{-1}}\right)\left(\frac{\partial\Sigma_{kl}^{-1}}{\partial\Phi_{ij}}\right) = \sum_{kl}\left(\frac{\partial\mathcal{L}}{\partial\Sigma_{kl}^{-1}}\right)\left(\frac{1}{2}\delta_{ki}\delta_{lj} + \frac{1}{2}\delta_{kj}\delta_{li}\right) = \frac{1}{2}\left(\frac{\partial\mathcal{L}}{\partial\Sigma_{ij}^{-1}} + \frac{\partial\mathcal{L}}{\partial\Sigma_{ji}^{-1}}\right). \tag{21}$$

Next we examine where the gradients of the Lagrangian vanish:

$$0 = \frac{\partial \mathcal{L}}{\partial \boldsymbol{\mu}} \quad \Longrightarrow \quad 0 = \boldsymbol{\Sigma}^{-1}(\boldsymbol{\mu} - \boldsymbol{\mu_0}) - \boldsymbol{\Sigma}^{-1}\boldsymbol{\lambda} \quad \Longrightarrow \quad \boxed{\boldsymbol{\lambda} = \boldsymbol{\mu} - \boldsymbol{\mu}_0} \tag{22}$$

$$0 = \frac{\partial \mathcal{L}}{\partial \boldsymbol{\lambda}} \quad \Longrightarrow \quad 0 = \boldsymbol{g} - \boldsymbol{\Sigma}^{-1}(\boldsymbol{\mu} - \boldsymbol{\theta}) \quad \Longrightarrow \quad \boxed{\boldsymbol{\mu} - \boldsymbol{\theta} = \boldsymbol{\Sigma}\boldsymbol{g}} \tag{23}$$

$$0 = \frac{\partial \mathcal{L}}{\partial \boldsymbol{\Phi}} \quad \Longrightarrow \quad 0 = \boldsymbol{\Sigma}_0 - \boldsymbol{\Sigma} + (\boldsymbol{\mu} - \boldsymbol{\mu}_0)(\boldsymbol{\mu} - \boldsymbol{\mu}_0)^\top - \left[\boldsymbol{\lambda}(\boldsymbol{\mu} - \boldsymbol{\theta})^\top + (\boldsymbol{\mu} - \boldsymbol{\theta})\boldsymbol{\lambda}^\top\right], \tag{24}$$

$$\Longrightarrow \quad \boxed{\boldsymbol{\Sigma} = \boldsymbol{\Sigma}_0 + (\boldsymbol{\mu} - \boldsymbol{\mu}_0)(\boldsymbol{\mu} - \boldsymbol{\mu}_0)^\top - \boldsymbol{\lambda}(\boldsymbol{\mu} - \boldsymbol{\theta})^\top - (\boldsymbol{\mu} - \boldsymbol{\theta})\boldsymbol{\lambda}^\top} \tag{25}$$

We claim that these equations (though nonlinear) can be solved in closed form. The first step is to eliminate $\boldsymbol{\lambda}$ from eq. (25) using eq. (22). In this way we find

$$\begin{aligned}
\boldsymbol{\Sigma} &= \boldsymbol{\Sigma}_0 + (\boldsymbol{\mu} - \boldsymbol{\mu}_0)(\boldsymbol{\mu} - \boldsymbol{\mu}_0)^\top - (\boldsymbol{\mu} - \boldsymbol{\mu}_0)(\boldsymbol{\mu} - \boldsymbol{\theta})^\top - (\boldsymbol{\mu} - \boldsymbol{\theta})(\boldsymbol{\mu} - \boldsymbol{\mu}_0)^\top & (26) \\
&= \boldsymbol{\Sigma}_0 - \boldsymbol{\mu}\boldsymbol{\mu}^\top + \boldsymbol{\mu}\boldsymbol{\theta}^\top + \boldsymbol{\theta}\boldsymbol{\mu}^\top + \boldsymbol{\mu}_0\boldsymbol{\mu}_0^\top - \boldsymbol{\mu}_0\boldsymbol{\theta}^\top - \boldsymbol{\theta}\boldsymbol{\mu}_0^\top & (27) \\
&= \boldsymbol{\Sigma}_0 + (\boldsymbol{\mu}_0 - \boldsymbol{\theta})(\boldsymbol{\mu}_0 - \boldsymbol{\theta})^\top - (\boldsymbol{\mu} - \boldsymbol{\theta})(\boldsymbol{\mu} - \boldsymbol{\theta})^\top. & (28)
\end{aligned}$$

It is worth highlighting the form of this equation:

$$\boldsymbol{\Sigma} = \boldsymbol{\Sigma}_0 + (\boldsymbol{\mu}_0 - \boldsymbol{\theta})(\boldsymbol{\mu}_0 - \boldsymbol{\theta})^\top - (\boldsymbol{\mu} - \boldsymbol{\theta})(\boldsymbol{\mu} - \boldsymbol{\theta})^\top$$

This is a simple rank-two update for $\boldsymbol{\Sigma}$. Note that $\boldsymbol{\Sigma} = \boldsymbol{\Sigma_0}$ if $\boldsymbol{\mu} = \boldsymbol{\mu}_0$; also, the solution for $\boldsymbol{\Sigma}$ is determined by the solution for $\boldsymbol{\mu}$.

Ultimately we must solve for $\boldsymbol{\mu}$, but first it is useful to solve for the intermediate quantity $\boldsymbol{g}^\top\boldsymbol{\Sigma}\boldsymbol{g} > 0$. From eq. (28), we obtain

$$\boldsymbol{g}^\top\boldsymbol{\Sigma}\boldsymbol{g} = \boldsymbol{g}^\top\boldsymbol{\Sigma}_0\boldsymbol{g} + \left[(\boldsymbol{\mu}_0 - \boldsymbol{\theta})^\top\boldsymbol{g}\right]^2 - \left[(\boldsymbol{\mu} - \boldsymbol{\theta})^\top\boldsymbol{g}\right]^2, \tag{29}$$

and from eq. (23), we obtain

$$\boldsymbol{g}^\top\boldsymbol{\Sigma}\boldsymbol{g} = \boldsymbol{g}^\top\boldsymbol{\Sigma}_0\boldsymbol{g} + \left[(\boldsymbol{\mu}_0 - \boldsymbol{\theta})^\top\boldsymbol{g}\right]^2 - \left(\boldsymbol{g}^\top\boldsymbol{\Sigma}\boldsymbol{g}\right)^2. \tag{30}$$

As shorthand, let $\rho = \boldsymbol{g}^\top\boldsymbol{\Sigma}\boldsymbol{g}$. Then from eq. (30) we see that $\rho$ satisfies the quadratic equation

$$\rho(1+\rho) = \boldsymbol{g}^\top\boldsymbol{\Sigma}_0\boldsymbol{g} + \left[(\boldsymbol{\mu}_0 - \boldsymbol{\theta})^\top\boldsymbol{g}\right]^2.$$

Note that there are no unknowns on the right side of this equation. The correct solution is given by the positive root since $\rho = \boldsymbol{g}^\top\boldsymbol{\Sigma}\boldsymbol{g} > 0$. Also note that $\rho = (\boldsymbol{\mu} - \boldsymbol{\theta})^\top\boldsymbol{g}$ from eq. (23).

It is useful to define one final intermediate quantity before solving for $\boldsymbol{\mu}$. Let

$$\boldsymbol{\varepsilon}_0 = \boldsymbol{\Sigma}_0\boldsymbol{g} - \boldsymbol{\mu}_0 + \boldsymbol{\theta}.$$

Note that $\boldsymbol{\varepsilon}_0$ simply measures the degree to which the parameters of $q_0(\boldsymbol{\theta})$ violate the desired constraint $\nabla_{\boldsymbol{w}}\log q(\boldsymbol{\theta}) = \nabla_{\boldsymbol{w}}\log p(\boldsymbol{\theta}, \boldsymbol{y})$. Put another way, if $\boldsymbol{\varepsilon}_0 = \boldsymbol{0}$, then we have the trivial solution $\boldsymbol{\mu} = \boldsymbol{\mu}_0$ and $\boldsymbol{\Sigma} = \boldsymbol{\Sigma}_0$.

Now we have everything to express the solution for $\boldsymbol{\mu}$ in a highly intuitive form; in particular, it will be immediately evident that $\boldsymbol{\mu} \to \boldsymbol{\mu}_0$ as $\boldsymbol{\varepsilon}_0 \to \boldsymbol{0}$. Starting from eqs. (23) and (28), we find

$$\begin{aligned}
\boldsymbol{\mu} - \boldsymbol{\mu}_0 &= \boldsymbol{\theta} - \boldsymbol{\mu}_0 + \boldsymbol{\Sigma}\boldsymbol{g}, & (31) \\
&= \boldsymbol{\theta} - \boldsymbol{\mu}_0 + \left[\boldsymbol{\Sigma}_0 + (\boldsymbol{\mu}_0 - \boldsymbol{\theta})(\boldsymbol{\mu}_0 - \boldsymbol{\theta})^\top - (\boldsymbol{\mu} - \boldsymbol{\theta})(\boldsymbol{\mu} - \boldsymbol{\theta})^\top\right]\boldsymbol{g}, & (32) \\
&= \boldsymbol{\varepsilon}_0 + (\boldsymbol{\mu}_0 - \boldsymbol{\theta})(\boldsymbol{\mu}_0 - \boldsymbol{\theta})^\top\boldsymbol{g} - (\boldsymbol{\mu} - \boldsymbol{\theta})(\boldsymbol{\mu} - \boldsymbol{\theta})^\top\boldsymbol{g}, & (33) \\
&= \boldsymbol{\varepsilon}_0 + (\boldsymbol{\mu}_0 - \boldsymbol{\theta})(\boldsymbol{\mu}_0 - \boldsymbol{\theta})^\top\boldsymbol{g} - (\boldsymbol{\mu} - \boldsymbol{\mu}_0 + \boldsymbol{\mu}_0 - \boldsymbol{\theta})(\boldsymbol{\mu} - \boldsymbol{\theta})^\top\boldsymbol{g}, & (34) \\
&= \boldsymbol{\varepsilon}_0 - \rho(\boldsymbol{\mu} - \boldsymbol{\mu}_0) + (\boldsymbol{\mu}_0 - \boldsymbol{\theta})[(\boldsymbol{\mu}_0 - \boldsymbol{\theta}) - (\boldsymbol{\mu} - \boldsymbol{\theta})^\top]\boldsymbol{g}, & (35) \\
&= \boldsymbol{\varepsilon}_0 - \rho(\boldsymbol{\mu} - \boldsymbol{\mu}_0) + (\boldsymbol{\mu}_0 - \boldsymbol{\theta})(\boldsymbol{\mu}_0 - \boldsymbol{\mu})^\top\boldsymbol{g}, & (36) \\
&= \boldsymbol{\varepsilon}_0 - (\rho\mathbf{I} + (\boldsymbol{\mu}_0 - \boldsymbol{\theta})\boldsymbol{g}^\top)(\boldsymbol{\mu}_0 - \boldsymbol{\mu}). & (37)
\end{aligned}$$

Note what has happened here: eq. (32) is a system of *nonlinear* equations for $\boldsymbol{\mu}$, but in eq. (37), all the nonlinearity has been expressed in terms of $\rho$. Since $\rho$ can be determined via eq. (30), we arrive effectively at a system of *linear* equations for $\boldsymbol{\mu}$. Collecting terms, we obtain

$$\left[(1+\rho)\mathbf{I} + (\boldsymbol{\mu}_0 - \boldsymbol{\theta})\boldsymbol{g}^\top\right](\boldsymbol{\mu} - \boldsymbol{\mu}_0) = \boldsymbol{\varepsilon}_0. \tag{38}$$

We thus arrive at the closed-form update

$$\boldsymbol{\mu} = \boldsymbol{\mu}_0 + \left[(1+\rho)\mathbf{I} + (\boldsymbol{\mu}_0 - \boldsymbol{\theta})\boldsymbol{g}^\top\right]^{-1}\boldsymbol{\varepsilon_0} \tag{39}$$

It is evident from this update that $\boldsymbol{\mu} \to \boldsymbol{\mu}_0$ as $\boldsymbol{\varepsilon}_0 \to \mathbf{0}$. The matrix inverse in this update can also be computed efficiently from the Woodbury matrix identity.

In sum, the joint update for $\boldsymbol{\mu}$ and $\boldsymbol{\Sigma}$ can be efficiently computed as follows:

1. Set $\boldsymbol{g} = \nabla_{\boldsymbol{w}} \log p(\boldsymbol{\theta}, \boldsymbol{y})$ and $\boldsymbol{\varepsilon}_0 = \boldsymbol{\Sigma}_0\boldsymbol{g} - \boldsymbol{\mu}_0 + \boldsymbol{\theta}$.
2. Solve $\rho(1+\rho) = \boldsymbol{g}^\top\boldsymbol{\Sigma}_0\boldsymbol{g} + \left[(\boldsymbol{\mu}_0 - \boldsymbol{\theta})^\top\boldsymbol{g}\right]^2$ for $\rho > 0$.
3. Compute $\boldsymbol{\mu} = \boldsymbol{\mu}_0 + \left[(1+\rho)\mathbf{I} + (\boldsymbol{\mu}_0 - \boldsymbol{\theta})\boldsymbol{g}^\top\right]^{-1}\boldsymbol{\varepsilon}_0$.
4. Compute $\boldsymbol{\Sigma} = \boldsymbol{\Sigma}_0 + (\boldsymbol{\mu}_0 - \boldsymbol{\theta})(\boldsymbol{\mu}_0 - \boldsymbol{\theta})^\top - (\boldsymbol{\mu} - \boldsymbol{\theta})(\boldsymbol{\mu} - \boldsymbol{\theta})^\top$.

Note that it straightforward to solve for the value of $\rho$ in step 2 of this procedure. In particular, from the quadratic formula, we obtain the positive root

$$\rho = \frac{\sqrt{1 + 4(\boldsymbol{g}^\top\boldsymbol{\Sigma}_0\boldsymbol{g} + \left[(\boldsymbol{\mu}_0 - \boldsymbol{\theta})^\top\boldsymbol{g}\right]^2)} - 1}{2}.$$

We can solve the linear equation for $\mu$ in step 3 of this procedure most efficiently by using the Sherman-Morrison formula

$$\left[a\mathbf{I} + \boldsymbol{u}\boldsymbol{g}^\top\right]^{-1} = \frac{1}{a}\left(\mathbf{I} - \frac{\boldsymbol{u}\boldsymbol{g}^\top}{a + \boldsymbol{u}^\top\boldsymbol{g}}\right), \quad \text{for every } \boldsymbol{u}, \boldsymbol{g}, a.$$

Applying this formula in step 3, we find that

$$\boldsymbol{\mu} = \boldsymbol{\mu}_0 + \frac{1}{1+\rho}\left[\mathbf{I} - \frac{(\boldsymbol{\mu}_0 - \boldsymbol{\theta})\boldsymbol{g}^\top}{1 + \rho + (\boldsymbol{\mu}_0 - \boldsymbol{\theta})^\top\boldsymbol{g}}\right]\boldsymbol{\varepsilon_0}. \tag{40}$$

Next we relate this update to the difference in the scores of the target distribution and the current estimate of the variational approximation. Since $\nabla_{\boldsymbol{\theta}} \log q_0(\boldsymbol{\theta}) = -\boldsymbol{\Sigma}_0^{-1}(\boldsymbol{\theta} - \boldsymbol{\mu}_0)$, we have that

$$\boldsymbol{\varepsilon_0} = \boldsymbol{\Sigma}_0\left(\boldsymbol{g} - \boldsymbol{\Sigma}_0^{-1}(\boldsymbol{\mu}_0 + \boldsymbol{\theta})\right) = \boldsymbol{\Sigma}_0\left(\boldsymbol{g} - \nabla_{\boldsymbol{\theta}} \log q_0(\boldsymbol{\theta})\right).$$

It follows, by substituting the above expression into Eq. 40, that

$$\boldsymbol{\mu} = \boldsymbol{\mu}_0 + \frac{1}{1+\rho}\left[\mathbf{I} - \frac{(\boldsymbol{\mu}_0 - \boldsymbol{\theta})\boldsymbol{g}^\top}{1 + \rho + (\boldsymbol{\mu}_0 - \boldsymbol{\theta})^\top\boldsymbol{g}}\right]\boldsymbol{\Sigma}_0\left(\nabla_{\boldsymbol{\theta}} \log p(\boldsymbol{x}, \boldsymbol{\theta}) - \nabla_{\boldsymbol{\theta}} \log q_0(\boldsymbol{\theta})\right),$$

which gives the update in the statement of the theorem.

Finally we prove that if $\boldsymbol{\Sigma_0}$ is positive definite, then so is $\boldsymbol{\Sigma}$. To do so, we show explicitly that all of the eigenvalues of $\boldsymbol{\Sigma}$ are positive. We begin by rewriting our results for $\boldsymbol{\Sigma}$ in eq. (28) and $\rho$ in eq. (30) in a more convenient form. As shorthand, let

$$\mathbf{M}_0 = \boldsymbol{\Sigma}_0 + (\boldsymbol{\mu}_0 - \boldsymbol{\theta})(\boldsymbol{\mu}_0 - \boldsymbol{\theta})^\top, \tag{41}$$

so that $\mathbf{M}_0$ captures the first two terms on the right side of eq. (28). Note that $\mathbf{M}_0$ is positive-definite, a fact that we will exploit repeatedly in what follows. In addition, recall that $\boldsymbol{\mu} - \boldsymbol{\theta} = \boldsymbol{\Sigma}\boldsymbol{g}$ from eq. (23). Thus with this notation we can rewrite eqs. (28) and (30) as

$$\boldsymbol{\Sigma} = \mathbf{M}_0 - (\boldsymbol{\Sigma}\boldsymbol{g})(\boldsymbol{\Sigma}\boldsymbol{g})^\top, \tag{42}$$
$$\rho(1+\rho) = \boldsymbol{g}^\top\mathbf{M}_0\boldsymbol{g}. \tag{43}$$

Now let $e$ be any normalized eigenvector of $\mathbf{\Sigma}$; we want to show that its corresponding eigenvalue $\lambda_e$ is positive. From eq. (42), it follows that

$$
\begin{align}
\lambda_e &= e^\top \mathbf{\Sigma} e \tag{44} \\
&= e^\top \left[ \mathbf{M}_0 - (\mathbf{\Sigma} g)(\mathbf{\Sigma} g)^\top \right] e \tag{45} \\
&= e^\top \mathbf{M}_0 e - \lambda_e^2 (e^\top g)^2. \tag{46}
\end{align}
$$

Note that if $e^\top g = 0$, then it follows trivially that $\lambda_e = e^\top \mathbf{M}_0 e > 0$. So we only need to consider the non-trivial case $e^\top g \neq 0$. To proceed, we note that

$$
(e^\top \mathbf{M}_0 g)^2 = (e^\top \mathbf{M}_0^{\frac{1}{2}} \mathbf{M}_0^{\frac{1}{2}} g)^2 \leq (e^\top \mathbf{M}_0 e)(g^\top \mathbf{M}_0 g), \tag{47}
$$

where we have used the Cauchy-Schwartz inequality to bound $(e^\top \mathbf{M}_o g)$ in terms of $(e^\top \mathbf{M}_0 e)$, the latter of which appears in eq. (46). Substituting this inequality into eq. (46), we find that

$$
\lambda_e \geq \frac{(e^\top \mathbf{M}_0 g)^2}{g^\top \mathbf{M}_0 g} - \lambda_e^2 (e^\top g)^2. \tag{48}
$$

To prove that $\lambda_e > 0$ we need one more intermediate result. Focusing on the rightmost term in this equality, we note that

$$
\lambda_e(e^\top g) = e^\top \mathbf{\Sigma} g = e^\top \left[ \mathbf{M}_0 - (\mathbf{\Sigma} g)(\mathbf{\Sigma} g)^\top \right] g = e^\top \mathbf{M}_0 g - \lambda_e(e^\top g)(g^\top \mathbf{\Sigma} g), \tag{49}
$$

and rearranging the terms in this equation, we find

$$
e^\top \mathbf{M}_0 g = \lambda_e(e^\top g)(1 + g^\top \mathbf{\Sigma} g). \tag{50}
$$

This intermediate result is useful because it relates the two terms on the right side of eq. (48). In particular, using eq. (50) to eliminate the term $e^\top \mathbf{M}_0 g$ in eq. (48), we find:

$$
\begin{align}
\lambda_e &\geq \frac{\left[ \lambda_e(e^\top g)(1 + g^\top \mathbf{\Sigma} g) \right]^2}{g^\top \mathbf{M}_0 g} - \lambda_e^2 (e^\top g)^2 \\
&= \lambda_e^2 (e^\top g)^2 \left[ \frac{(1 + g^\top \mathbf{\Sigma} g)^2}{g^\top \mathbf{M}_0 g} - 1 \right] \\
&= \lambda_e^2 (e^\top g)^2 \left[ \frac{(1 + \rho)^2}{\rho(1 + \rho)} - 1 \right] \\
&= \frac{\lambda_e^2 (e^\top g)^2}{\rho}, \\
&> 0,
\end{align}
$$

where the final inequality follows because the individual terms $\lambda_e^2$, $(e^\top g)^2$, and $\rho$ are all strictly positive; note that $\lambda_e$ cannot be equal to zero because this contradicts the equality in eq. (46). This completes the proof. Perhaps it is useful that this derivation also gives upper bounds on $\lambda_e$, namely

$$
\frac{1}{\lambda_e} \geq \frac{(e^\top g)^2}{\rho} \implies \lambda_e \leq \frac{\rho}{(e^\top g)^2} = \frac{g^\top \mathbf{\Sigma} g}{(e^\top g)^2}. \tag{51}
$$

$\square$

