# OpenReview forum: "Variational Inference with Gaussian Score Matching"
_NeurIPS.cc/2023/Conference — NeurIPS 2023 poster_

### Official Review · Reviewer_CVEd · 2023-06-22

**Soundness:** 2 fair
**Presentation:** 4 excellent
**Contribution:** 2 fair
**Rating:** 3
**Confidence:** 5

**Summary:**

The submission proposes a method for black-box variational inference based that is based on score matching. The method follows an iterative procedure, where at every iteration the variational approximation is updated by first obtaining a single sample and then updating the approximation such that the gradient of the log-densities of the approximation matches the gradient the gradient of the target distribution on that sample. As in general, several solutions might to this constrained problem might exist, the proposed method aims to also minimize the forward KL divergence to the approximation of the previous iteration.

The paper shows that the optimal parameters are a stationary point, if the optimal distribution can match the score everywhere. This can be easily seen from the constrained optimization problem, as the last approximation minimizes the KL to the last approximation and also satisfies the constraints.

The constrained optimization problem (minimizing the forward KL to the previous distribution, while matching the gradients of the target) can in general not be solved in closed form, and the submission does not discuss how the algorithm can be implemented in the general setting, nor does it perform any evaluations. Instead, the work focuses on Gaussian variational approximations, and shows that in this (quite relevant) special case, the constrained optimization problem can be solved in closed form to provide a hyperparameter-free update.

The method is evaluated on several test problems and compared to a baseline which maximizes the ELBO using the reparameterization trick. The proposed method converges around two orders of magnitude faster than using the reparameterization trick and also often learns better or similar approximations. However, in particular for target distributions that are very different from Gaussian distributions, the methods suffers from high oscillations and does not seem to converge.


**Strengths:**

Originality
-------------
The proposed method seems to be novel, and performing variational inference without explicitly minimizing a particular divergence could be interesting.

Clarity
--------
The method is described perfectly clear. The paper was very to read and follow. I also like that the limitations (the method may not converge when the target distribution is more expressive than the variational approximation, the method does not minimize the KL or any other divergence) is clearly communicated.

Relevance
--------------
Gaussian variational inference is an important problem setting with various applications.

**Weaknesses:**

Technical Soundness
-----------------------------
While the paper shows that the optimal parameters are a stationary point if the target distribution can be perfectly approximated, it does not provide any analysis for the setting that the approximation can not perfectly approximate the target distribution. Given that the paper focuses on Gaussian variational approximations, this lack of analysis is a major shortcoming. Of course, without specifying a particular divergence it is not even clear which parameter would be "optimal" in the setting where the target distribution can be perfectly matched. However, it would be important to better analyze, if there are any mild conditions for convergence, and which criteria the learned approximation fulfills (is there any---potentially obscure---divergence that is minimized?). Further, even in the setting where the target distribution can be perfectly approximated, technically the paper only shows that the optimal approximation is a stationary point, but it does not prove that the method actually converges to that point or that there are no other stationary points. Actually, both can be shown straightforwardly from the fact that a Gaussian distribution has full support, but the current submission does not make this point, and, furthermore, the sample complexity might nevertheless be quite bad.

Related Work
------------------
The paper focuses on Gaussian variational inference, but misses the most relevant works in that area. Natural gradient based methods are known to significantly outperform methods based on the vanilla gradient (such as the BBVI/Reparameterization-Trick baseline used in the paper).
For Gaussian variational approximations, the natural gradient can be approximated extremely efficient (almost as fast as the vanilla gradient) and enjoys faster convergence and empirically also better exploration. There is broad literature of natural gradient descent for Gaussian variational approximations based on zero-order, first-order and second-order information:

- VIPS (Arenz et al.. 2020) uses a modified version of a policy search method (called MORE), which is based on compatible function approximation. This approach use least-squares based on samples from the current approximation to fit a quadratic model to the target distribution (therefore, it only require function evaluations but no differentiable target distributions). It can be shown that the parameters of this quadratic model can be used to compute the natural gradient (also in closed form). Their work focuses on GMM approximation, but it uses independent updates to the Gaussian components.

- Lin et al. (2019) show how first order information can be used for approximating the natural gradient. They also consider the GMM setting, but also use independent update for the Gaussian components. Their method estimates the natural gradient of a Gaussian variational distribution using Stein's Lemma. Their estimate of the natural gradient is a simple linear function of the gradient of the log-ratio (between target and approximation) evaluated on samples from the approximation.

- VOGN (Khan & Nielsen, 2018) estimates the natural gradient using both gradients and Hessians of the joint distribution.

Furthermore, there are also other methods (not based on natural gradients) that can be used for optimizing Gaussian variational approximations. For example, when the Hessian of the target distribution is available, HFSGVI (Fan et al., 2015) or TRUST-VI (Regier et al. 2017) can be applied.


Evaluation
--------------
The evaluation is insufficient as it lacks the most important baselines VIPS and the method by Lin et al. (2019). Both methods are applicable to the problem setting of the submission, and as shown by Arenz et al. (2020, see Appendix K) even the zero-order method significantly outperforms the reparameterization-baseline used in the submission.

Furthermore, the performance of the baseline seems to be worse than it should. In the experiments of Section 3.1, where a Gaussian distribution is used as target distribution, the paper states that the reparameterization trick converges to suboptimal solutions. However, the ELBO between two Gaussians has a single stationary point which is the global minimizer. Hence, the reparameterization trick should not converge to a suboptimal solution. I can imagine that the vanilla gradient can lead to crawling, however, when using ADAM this should actually not happen as momentum should build up. I suspect that either the hyperparameters are badly chosen (too large stepsizes may lead to oscillations that prevent ELBO improvements) or problems with the implementation (numerical errors when computing the log-density of the Gaussians).

Minor Comment
---------------------
"In the experiments with synthetic models in Sections 3, 3.1, and 3.1 [sic]"

References
---------------
Arenz, Oleg, Mingjun Zhong, and Gerhard Neumann. "Trust-region variational inference with gaussian mixture models." The Journal of Machine Learning Research 21.1 (2020): 6534-6593.

Lin, W., Khan, M. E., & Schmidt, M. (2019). Stein's Lemma for the Reparameterization Trick with Exponential Family Mixtures. arXiv preprint arXiv:1910.13398.

Mohammad Emtiyaz Khan and Didrik Nielsen. Fast yet simple natural-gradient descent for variational inference in complex models. In 2018 International Symposium on Information Theory and Its Applications (ISITA), pp. 31–35. IEEE, 2018.

K. Fan, Z. Wang, J. Beck, J. T. Kwok, and K. Heller. Fast second-order stochastic backpropagation for variational inference. In Advances in Neural Information Processing
Systems, NIPS’15, pages 1387–1395, 2015.

J. Regier, M. I. Jordan, and J. McAuliffe. Fast black-box variational inference through stochastic trust-region optimization. In Advances in Neural Information Processing
Systems 30, pages 2399–2408, 2017.

**Questions:**

Why does the reparameterization trick converge to suboptimal solutions when the target distribution is Gaussian? Can you provide code and additional details on hyperparameter tuning? Can you provide an explanation for this surprising result?

For changing my opinion, the paper also needs to discuss the relevant methods in this problem setting (above references), and ideally add a comparison to the work by Lin et al. (2019) or Arenz et al. (2020).

**Limitations:**

The approach does have serious limitations, mainly by not proving convergence. However, I already stated the limitations under weaknesses and I also think that the limitations are adequately discussed in the paper.

---

> ### Author Rebuttal · Authors · 2023-08-09
>
> >It would be important to better analyze, if there are any mild conditions for convergence, and which criteria the learned approximation fulfills.
>
> We agree with the referee that its important, but challenging to analyze given that GSM does not explicitly minimize any divergence. Empirically, we have so far observed that GSM converges to solutions that have the best forward KL divergence (and similar reverse KL divergence as BBVI for non-Gaussian target). Furthermore, as far as we know, there are no theoretical convergence guarantees for BBVI when the variational family is misspecified either. It is a difficult open question still under active research years after the algorithm has been widely used.
>
> > In the setting where the target distribution can be perfectly approximated, ..., the method actually converges to that point or that there are no other stationary points. Actually, both can be shown straightforwardly from the fact that a Gaussian distribution has full support, but the current submission does not make this point
>
> We thank the referee for making this point. We have now updated Lemma 1 to clarify this.
>
> > Natural gradient based methods are known to significantly outperform methods based on the vanilla gradient (such as the BBVI). For changing my opinion, the paper also needs to discuss the relevant methods and ideally add a comparison to the work by Lin et al. 2019 or Arenz et al. 2020.
>
> We thank the referee for pointing to natural gradient descent based VI as a baseline. We will certainly discuss these works and add this  baseline for comparison in the revision. As requested, we have added a comparison to NGD-VI based on Lin et.al. for a Gaussian target in Fig. 2 in the uploaded pdf. However, note that for full rank Gaussians, NGD has cubic scaling $d^3$ with dimension while GSM has quadratic $d^2$. We now summarize our implementation & results (also see the general answer above)
>
> *Implementation*
> - We use only the first order information for fair comparison to GSM.
> - The updates for variational parameters $\mu_i$ and $\Sigma_i$ follow Eq. 16 of Lin et.al (arxiv:1906.02914) adapted for a single Gaussian instead of GMM
> - Eq 16 requires the Hessian of the ELBO. To approximate this with 1st-order information, we implement VOGN update (Eq. 10 of Khan et.al, arXiv:1806.04854). This uses Gauss-Newton approximation for the Hessian of  the likelihood $\log p(\theta | x)$ and $\Sigma_i^{-1}$ for Hessian of $\log q$.
>
> *Results*
>
> Fig. 2 of attached shows results for NGD-VI for a D=64 Gaussian target by varying the hyperparameters- learning rate and batch size. We find that
> - As the referee pointed out, NGD-VI significantly outperforms BBVI.
> - “Optimally tuned" NGD-VI can be competitive with GSM.
> - However the performance of NGD-VI is very sensitive to the hyper-parameters. Larger batch sizes allow us to use larger learning rate so they need to be tuned together. However a large step size gets stuck oscillating around the minima and hence we need to additionally tune the scheduling of the learning rate.
> - We couldn't experiment with optimizers like Vprop/Vadam from Khan et.al. for the rebuttal but will investigate those for the updated manuscript.
> - For $D\geq256$, NGD-VI was prone to diverging for $lr>0.01$. To ensure convergence, we added an additional regularization parameter for Hessian estimates in VOGN.
>
> *Compared to GSM*
> 1) For VI with a full-rank Gaussian, GSM has a quadratic complexity of $d^2$ while NGD-VI is cubic $d^3$ as it requires estimating both $\Sigma$ and $\Sigma^{-1}$ for each iteration.
> 2) An optimally tuned NGD-VI can be competitive with GSM, but this will require tuning the learning rate, it's scheduling and batch size. In contrast, GSM has no learning rate parameters to tune, and is largely insensitive to batch size.
> 3) For large problems, we need a regularization hyperparameter for Hessian in VOGN updates. GSM has no such issues in scaling.
>
> Thus based on our experiments, we argue that as compared to NGD-VI, GSM is faster in terms of gradient evaluations, has a smaller iteration complexity, and is easy to tune.
>
> However we agree with the referee that it was a relevant baseline missing from the draft and would like to thank them for pointing this.
>
> > BBVI should not converge to a suboptimal solution for Gaussians. Can you provide code and details on hyperparameter tuning?
>
> We agree that BBVI will converge for Gaussian targets, and have now uploaded additional figures to show this. In particular, BBVI with a well chosen learning rate and batch size, will converge given a sufficient number of iterations. For instance, in Fig. 1 of uploaded pdf, we show the performance of BBVI for a D=10 Gaussian target for different values of the hyperparameters- learning rate $lr$ & batch size $B$. In this case, BBVI for $B\geq4$ & $lr \leq 0.01$ eventually does converge to the same solution as GSM in terms of reverse KL (right panel, blue & orange lines). We hope that this resolves the misunderstanding- we are not claiming that BBVI *never* converges to the correct solution for the Gaussian target. Instead, it did not converge under our finite budget. With more iterations BBVI eventually converges.
>
> *Hyperparameters*- We fixed $B=2$ and then tuned $lr$ with a grid-search, but we didn't tune it's scheduling. As seen in Fig 1, while larger step-sizes converge faster at the beginning, they only converge in a small radius around the minima. Thus one also needs to tune the scheduling of step size. In Fig.1 attached, we now also tune the batch size $B$ of BBVI and find that GSM still outperforms BBVI. We will include this discussion and figure in revision to give more context to the results of BBVI.
>
> We have provided our code with the initial submission in the supplementary. We will make a package public with all 3 algorithms after the review process.

---

> > ### Comment · Reviewer_CVEd · 2023-08-16
> > **Baseline Comparisons still suboptimal**
> >
> > Thank you very much for the response and the additional experiments. However, the choice of using VOGN for estimating the expected Hessian is quite suboptimal, because it only provides a biased estimate. Instead, Stein's Lemma ( see the citation to Lin et al. above) provides efficient unbiased estimates of the expected Hessian while just relying on first-order information.
> >
> > I quickly tested a recent implementation of NGVI with Stein's Lemma for GMMs (using the special case of using one component). And I obtained an ELBO of 0 for the frgaussian.py task after 25 samples.
> >
> > To reproduce, you can extend the runs.py with the code block below. The new method run_gmmvi can be called from frgaussian.py (likely also for the other environments) in the same way as gsm /bbvi. However the class does not save results and log the output in the same way, but just uses the prints from the used framework (https://gmmvi.rtfd.io). The code is based on an example script provided at gmmvi.rtfd.io
> > It only takes one parameter as input, which is --batch for specifying the maximal number of samples to be drawn per iteration.
> >
> > The code should run after `pip install gmmvi`. I did not make any more evaluations, so it would be interesting to compare the performance on the whole test suite, also with respect to other divergences (e.g. forward kl).
> >
> > I encourage the authors to improve the experiment section. Personally, I would not oppose publication of the work despite the limitations that I raised in my original review, as long as the limitations are well discussed (so far the submission did make a good job in that respect!). I do think that the submission provides interesting novel ideas. However, I think that currently the empirical performance compared to strong baselines is still overstated.
> >
> > ```
> > from gmmvi.gmmvi_runner import GmmviRunner
> > from gmmvi.configs import get_default_algorithm_config, update_config
> >
> > def run_gmmvi(args, model, x0, modelpath, callback, samples):
> >     '''This function is based on the following example https://gmmvi.readthedocs.io/en/latest/get_started.html#using-the-gmmvirunner-with-custom-environments
> >     '''
> >
> > # For creating a custom environment, we need to extend
> > # gmmvi.experiments.target_distributions.lnpdf.LNPDF:
> >     from gmmvi.experiments.target_distributions.lnpdf import LNPDF
> >     class Target(LNPDF):
> >         def __init__(self):
> >             super(Target, self).__init__(safe_for_tf_graph=False)
> >             self.model = model
> >
> >         def get_num_dimensions(self) -> int:
> >             return self.model.d
> >
> >         def log_density(self, samples: tf.Tensor) -> tf.Tensor:
> >             return self.model.log_prob(samples)
> >
> >
> >     # We can also use the GmmviRunner, when using custom environments, but we have
> >     # to put the LNPDF object into the dict. Furthermore, we need to define the other
> >     # environment-specific settings that would otherwise be defined in
> >     # the corresponding config in gmmvi/config/experiment_configs:
> >     environment_config = {
> >         "target_fn": None, ## I somehow couldn't add Target() here, but I had to add it after merge_configs()
> >         "start_seed": 0,
> >         "environment_name": "GSMTARGET",
> >         "model_initialization": {
> >             "use_diagonal_covs": False,
> >             "num_initial_components": 1,
> >             # Does GSM/BBVI use the same initial Gaussian???
> >             "prior_mean": 0.,
> >             "prior_scale": 1.,
> >             "initial_cov": 1.,
> >         },
> >         "gmmvi_runner_config": {
> >             "log_metrics_interval": 1
> >         },
> >         "use_sample_database": True,
> >         "max_database_size": int(1e6),
> >         "temperature": 1.
> >     }
> >
> >     algorithm_config = get_default_algorithm_config("SEMTRON") # The recommended variant is SAMTRON, SEMTRON does not add additional components
> >     algorithm_config['sample_selector_config']['desired_samples_per_component']=args.batch # Only hyperparameter worth tuning?
> >
> >     # Now we just need to merge the configs and use GmmviRunner as before:
> >     merged_config = update_config(algorithm_config, environment_config)
> >     merged_config['target_fn']=Target()
> >     gmmvi_runner = GmmviRunner.build_from_config(merged_config)
> >
> >     for epoch in range(100):
> >         gmmvi_runner.iterate_and_log(epoch)
> >
> >     return None, None, None, None
> > ```

---

> > > ### Author Response · Authors · 2023-08-17
> > > **New experiments with GMMVI**
> > >
> > > We thank the reviewer for providing a detailed code, especially the one that followed the format of our own submitted code. We were able to install the `gmmvi`  package and use the code block to run the experiments along the lines you suggest.
> > >
> > > The only parameter that we changed in the code is the batch size, as that is the only free parameter in our GSM method. (Note this means we did not tune the learning rate or its scheduling, but used the default settings.)
> > >
> > > > I quickly tested a recent implementation of NGVI with Stein's Lemma for GMMs (using the special case of using one component). And I obtained an ELBO of 0 for the frgaussian.py task after 25 samples.
> > >
> > > We assume this is for the experiment with default parameters in frgaussian.py submitted in the code. These correspond to a Gaussian target with $D=4$. For this sized problem, we are able to exactly replicate the referee's result with `gmmvi`. However in this example, our GSM-VI method also takes only ~20 iterations.
> > >
> > > We then experimented with changing $D$ (the size of the problem) to $D=10, 20$. We found that GSM significantly outperforms the NGD implementation of the code. Specifically NGD's performance is sensitive to the batch size and initialization. Further we found that some of the runs of NGD diverged. This was mostly driven by the fact that, as we increase dimensions, the Hessian approximation of NGD becomes singular. We have contacted the AC to see if we can share a new figure summarizing these results through them.
> > >
> > > > I encourage the authors to improve the experiment section.
> > >
> > > We will be happy to include results from NGD-VI, both from our own implementation and gmmvi package, in the revision of our work.
> > >
> > > >  However, I think that currently the empirical performance compared to strong baselines is still overstated.
> > >
> > > In our revision, we agree to make our claims and statements regarding baselines more precise. Though note, we still found that GSM-VI consistently outperformed NGD-VI (with both Gaus-Newton and Stein's Hessian approximation) in our experiments. It's true that, when properly tuned, NGD-VI can be competitive with GSM. But we found it is very sensitive to tuning parameters and is harder to scale on account of its approximating the Hessian with first order information. Furthermore, the method remains cubic in computational complexity, while GSM is quadratic.

---

> > > > ### Comment · Reviewer_CVEd · 2023-08-18
> > > > **Re: New experiments with GMMVI**
> > > >
> > > > Thank you for your first evaluations.
> > > >
> > > > **Regarding numerical instabilities** When running NGVI with very small batch-sizes of 2, instabilities due to bad NG estimates can be expected. However, when using more reasonable batch sizes, for example when using the default stepsize by uncommenting the corresponding line in the run_gmmvi(), I did not observe instabilities even for $D=100$.
> > > >
> > > > **Tuning of hyperparameters** I agree that not relying on tuning a hyper-parameter is an advantage of GSM-VI. Still, at least corse hyperparameter tuning should be performed for the baseline when evaluating their performance. It also seems like the batch-size is rather easy to tune, because setting a larger batch-size generally leads to better stability, while reducing the sample efficiency. Hence, one just needs to run a handful of runs (or a small linesearch) over the batchsize to find the smallest batchsize that still achieves a good performance.
> > > >
> > > > **Computational efficiency** Why is the complexity in $\mathcal{O}(n^2)$? Judging from the code, GSM-VI also uses $\mathbf{\Sigma}$ and $\mathbf{\Sigma}^{-1}$ for sampling and evaluating the gradient respectively. Furthermore, for realistic problems, evaluating the target densitiy becomes the bottleneck, not the evaluations of the Gaussian approximation. Also, the required computational time seems to be significantly larger due to the fact that GSM-VI does not benefit from larger stepsizes, and therefore needs to perform much more iteration while being less able to benefit from parallelization.
> > > >
> > > > Anyways, the sample efficiency of GSM-VI seems to be comparable to NGVI and maybe for some Gaussian-like targets even slightly better.

---

> > > > > ### Author Response · Authors · 2023-08-18
> > > > > **Re: Re: New experiments with GMMVI**
> > > > >
> > > > > Thank you for the response.
> > > > >
> > > > > > Regarding numerical instabilities and Tuning of hyperparameters:
> > > > >
> > > > > Our experiment above was done with different batch sizes for NGD-VI with gmmvi package (batch sizes 1, 2, 4, 8, which seamed reasonable for D=20). Furthermore, for larger batch sizes, we did not find that NGVI necessarily became more stable. For example, with D= 20, for different batch sizes, and a budget of 500 gradient evaluations, the resulting reverse KL divergence for NGD-VI, and for GSM, was
> > > > >
> > > > > | NGD B=1   | NGD B=2         | NGD B=4       | NGD B=8          | GSM B=1   |
> > > > > | --- | --- | --- | --- | --- |
> > > > > | diverged | $10^0$       | $10^4$       | $10^2$          | $10^{-5}$ |
> > > > >
> > > > > Thus even tuning the batch size, in larger dimensions GSM significantly outperformed this implementation NGVI.
> > > > >
> > > > > >  " when using the default stepsize by uncommenting the corresponding line in the run_gmmvi(), ".
> > > > >
> > > > > We do not follow which line in the code the reviewer is referring to in their statement - as mentioned previously, we used the shared code block as is except changing the batch size argument and hence believe the code is using the default settings for other parameters.
> > > > >
> > > > > >  Computational efficiency Why is the complexity in $\mathcal{O}(n^2)$? Judging from the code, GSM-VI also uses $\mathbf{\Sigma}$ and $\mathbf{\Sigma}^{-1}$ for sampling and evaluating the gradient respectively.
> > > > >
> > > > > GSM does not use $\Sigma^{-1}$ at any stage of the algorithm. This can be seen most straightforwardly in Algorithm 1 in our paper. Though the gradient of $\log q$ does appear in Eq. 5 of the paper, which itself relies on $\Sigma^{-1}$ , it is never evaluated explicitly. This is because it gets multiplied out by $\Sigma$ in the $A$ term in Eq. 5. In the submitted code, the GSM updates are evaluated in the function ```src/gsm/gaussian_update_batch```  which only has matrix-vector products implemented with einsum and operations over $\Sigma$. There is no $\Sigma^{-1}$ anywhere in our code.
> > > > >
> > > > > > Also, the required computational time seems to be significantly larger due to the fact that GSM-VI does not benefit from larger stepsizes,
> > > > >
> > > > > Here we respectfully disagree. First, we found in our experiments GSM required fewer gradient evaluations, and each iteration of GSM is cheaper due to aforementioned quadratic computational complexity. Second, on your perceived disadvantage of not having a stepsize, note that GSM also takes larger stepsizes earlier on, and smaller stepsizes towards the end. This is because the updates are proportional to the residual of the score matching equation, see for instance eq (5). As the model becomes a better fit, the residual decreases, and so do the magnitude of the updates. This is how and why GSM adapts without a stepsize scheduler. Finally, not needing to tune a stepsize is a significant practical advantage. Tuning a stepsize costs additional resources.
> > > > >
> > > > > > and therefore needs to perform much more iteration while being less able to benefit from parallelization.
> > > > >
> > > > > Here we did not understand the reviewers comment on parallelization. Note that GSM allows for batch updates, which can be done in parallel. If this question is in reference to our recommendation of using batch size B=1 or 2, then there seems to be a misunderstanding. We state that B=1 or larger perform equally well in terms the total number of gradient evaluations. However in terms of wall-clock time, if we can parallelize the batch updates, then the larger batch size will certainly provide additional gains.
> > > > >
> > > > > >  Furthermore, for realistic problems, evaluating the target densitiy becomes the bottleneck, not the evaluations of the Gaussian approximation.
> > > > >
> > > > > Here we agree with the reviewer, and this is why we evaluated performance with regards to the number of gradient evaluations of the target density.

---

> > > > > > ### Comment · Reviewer_CVEd · 2023-08-18
> > > > > > **Tuning the batchsize**
> > > > > >
> > > > > > > Thus even tuning the batch size, in larger dimensions GSM significantly outperformed this implementation NGVI.
> > > > > >
> > > > > > As I said before, the tested  batchsizes are simply too small. With a batchsize of 30 I also get an ELBO of 0.0000 with a budget of 500 function evaluations (running the following command: python3 -u frgaussian.py -D 20 -r 4  --batch 30 (Note that this batchsize was not tuned, but was a first guess)
> > > > > >
> > > > > > > We do not follow which line in the code the reviewer is referring to in their statement
> > > > > >
> > > > > > I refer to the line that overwrites the default batchsize using the --batch parameter: `    algorithm_config['sample_selector_config']['desired_samples_per_component']=args.batch # Only hyperparameter worth tuning?`
> > > > > >
> > > > > > > Here we did not understand the reviewers comment on parallelization.
> > > > > >
> > > > > > While GSM can use larger batch sizes, there seems to be rather small benefit in increasing it (in terms of reducing the number of iterations to converge). Hence, the fastest way to converge is using  small batch-sizes (eg. 2). However, 100 iterations with batchsize 2, produce much more computational overhead compared to 2 iterations with batchsize 100 (which can make much better use of parallelization). Hence, GSM evaluations take  significantly longer (at least on my machine).

---

> > > > > > > ### Author Response · Authors · 2023-08-19
> > > > > > > **Experiments with larger batch and parallelization**
> > > > > > >
> > > > > > > We thank the referee for improving our understanding of batch sizes expected in their code. We have now run this implementation of NGD-VI with gmm-vi package for $D=20, \ r=4 \, B=30$
> > > > > > >
> > > > > > > And we confirm that it’s performance has improved in terms of gradient evaluations. Here is our updated table with more runs:
> > > > > > >
> > > > > > > | NGD B=1 | NGD B=2   | NGD B=4  | NGD B=8   |  NGD B=15 | NGD B=20 | NGD B=30 | GSM B=1 |
> > > > > > > | --- | ---  | ---  | ---  |  --- | --- | --- | --- |
> > > > > > > | diverged | $10^0$  | $10^4$  | $10^2$   |  $\lt 10^{-5}$ | $\lt 10^{-5}$ | $ \lt10^{-5}$ | $\lt 10^{-5}$ |
> > > > > > >
> > > > > > > We now also monitor the number of iterations to reach $10^{-2}$ reverse KL (this value is the amplitude of oscillations around minima for NGD-VI). This is a better representation of how fast the different configurations converge. These are the results for the experiment above.
> > > > > > >
> > > > > > > | NGD B=1  | NGD B=2     | NGD B=4    | NGD B=8     |   NGD B=15  | NGD B=20  | NGD B=30  | GSM B=1  |
> > > > > > > | --- | --- | --- | --- | --- | --- | --- | --- |
> > > > > > > | diverged | did not converge   | diverged   | diverged   |  420  | 400 | 480 | 250 |
> > > > > > >
> > > > > > > We repeated the exercise with a larger problem : $D = 256,\ r = 256$ (note that we construct covariance matrix as a diagonal plus rank $r$ matrix)
> > > > > > >
> > > > > > > | NGD B=100  | NGD B=150   | NGD B=200     | NGD B=300    | GSM B=2 |
> > > > > > > | --- | --- | --- | --- | --- |
> > > > > > > | diverged | 11000 | 13000 | 17000    |  3000 |
> > > > > > >
> > > > > > >
> > > > > > > We also now understand, and agree with the reviewer's comment about parallelism: here on this experiment with Gaussian targets, NGD-VI will gain more from parallelism than our GSM.
> > > > > > >
> > > > > > > Though note this is not necessarily true for non-Gaussian targets where we do observe  benefits of increasing the batch size for GSM.
> > > > > > > We will add these comments to our revision when adding results with NGD-VI as baseline.
> > > > > > >
> > > > > > >
> > > > > > > > GSM evaluations take significantly longer (at least on my machine).
> > > > > > >
> > > > > > > The GSM code submitted with the paper is for validation purpose only and has not been optimized in any way for speed. We now have a much faster Jax implementation now which will be made public upon the end of the revision process. With this code, a single iteration of GSM and BBVI are comparable in timing. For e.g., for a full rank Gaussian target with $D=2048$ using $B=1$ after JIT-compilation, GSM update takes 230 ms/iteration while BBVI takes 227 ms/iteration.
> > > > > > >
> > > > > > >
> > > > > > > We thank the reviewer for this code and the discussion.

---

### Official Review · Reviewer_DLrt · 2023-07-05

**Soundness:** 3 good
**Presentation:** 4 excellent
**Contribution:** 3 good
**Rating:** 7
**Confidence:** 3

**Summary:**

A new variational inference (VI) framework is presented by matching the score of the variational distribution, q, with the target posterior p. Specifically, the closed form score matching equations for the Gaussian variational family are derived and the resulting method is named Gaussian Score Matching VI (GSM-VI). GSM-VI shows favorable results in multiple instances when comparing the method performance to Black Box VI (BBVI) on a multivariate Normal target, a sinh-arcsinh Normal target and real data from posteriordb database.


**Strengths:**

A simple, yet demonstrably fruitful, and novel VI framework is presented. The paper is easy to digest; it follows a natural progression of ideas and theoretical results are completed with intuitive explanations. GSMVI shows faster convergence w.r.t. number gradient estimates for several models and datasets w.r.t. BBVI. Furthermore, the paper presents clear scenarios where GSMVI is favorable to BBVI, e.g., when the covariance matrix is ill-conditioned.

**Weaknesses:**

The paper presents a novel VI approach, but the only instantiation of the framework in the paper is restricted to the Gaussian variational family setting, without any non-trivial theoretical results beyond this setting. Where VI provides techniques to find the optimal q within a variational family in terms KL-divergence to p, it is not clear how well score-matching VI performs in this setting (except for when q is in the same family as p).

The paper lacks explanation or deeper analysis of the large amplitude oscillations presented in the experiments; The cause of variance in BBVI is well-known and therefore research can focus on taming it. I am not well-acquainted with the score matching literature, to me these oscillations require an explanation.

Only reporting the Forward KL (FKL) could be misleading when comparing GSM-VI to other VI methods which optimization is based on the reversed KL. Reporting only the FKL could favor diffuse variational posteriors over a mode-seeking peaked variational posteriors (which are notoriously produced by VI methods); however, it is not clear that the diffuse variational posterior is to be preferred over the peaked variational posterior.

**Questions:**

I don't concur with the use of the term "closed-form updates" of line 229 as GSM-VI relies on sampling to be able to evaluate those update equations. This introduces variance in the updates, as opposed to CAVI updates for e.g. the exponential family where closed-form variational parameter updates can be derived without the need of sampling.

I would be ready to raise my score if the rebuttal addresses the following points:
1. Nuancing the experiments on simulated data with a metric less disadvantageous of VI, e.g. reverse KL, for at least one data set.
2. Addressing the issue raised under weaknesses regarding oscillatory behavior.


**Limitations:**

Yes, limitations are addressed.

---

> ### Author Rebuttal · Authors · 2023-08-09
>
> > I don't concur with the use of the term "closed-form updates" of line 229 as GSM-VI
>
> By closed-form updates, we refer only to the fact that (4) has a closed form solution for the Gaussian variational family for a generated sample. That is, the explicit update equations are given in equations (5) and (6) (detailed in Theorem 2.2). By closed-form we do not mean that the updates in equations (5) and (6) are a solution to global score matching equations in (3). We will clarify this distinction in the revision.
>
> > I would be ready to raise my score if the rebuttal addresses the following points:
>
> We thank the reviewer for this consideration. We have now addressed both these concerns regarding additional metrics in the experiments, and the presence of oscillations.
>
> > Nuancing the experiments on simulated data with a metric less disadvantageous of VI, e.g. reverse KL, for at least one data set.
>
> For real-world problems from posteriodb models, we already show the results for reverse KL divergence.
>
> But we agree with the referee that even for synthetic models, showing the evolution of reverse KL i.e. the objective which is explicitly optimized by BBVI, in addition to forward KL can be instructive. We have now attached a version of this figure in the uploaded pdf  in the general response for a Gaussian target of 16 dimensions (see Figure 1). In this figure, we also present the results by varying hyperparameters of BBVI i.e the learning rate and batch size. This clearly shows that for Gaussian targets, BBVI can converge to the same quality solution as GSM but requires much larger batch size and small learning rates (only B>4, lr<0.01 converged in the 5000 iterations). Hence the conclusions remain unchanged from the Forward KL metrics presented in the original manuscript. Please see the attached pdf and the discussion in the  "Author Rebuttal" at the top for more details.
>
> We will add this discussion and results showing reverse KL for other examples in the supplementary material of the revised manuscript.
>
> > Addressing the issue raised under weaknesses regarding oscillatory behavior.
>
> Through additional experiments in the attached pdf, we believe we now have a better understanding of the oscillatory behavior that the referee points out, and we summarize this in the following points:
> - When the variational family is well specified, i.e the target distribution is Gaussian or can be fit by one, we find no oscillations in GSM for any batch size and the algorithm converges to an optimal solution.
> - When the target is non-Gaussian, the monitored KL divergence has oscillations. It turns out that this was because we were using batch size B=2 for GSM for all experiments. For larger batch sizes, these oscillations are suppressed very easily. We have attached a figure (see Figure 3) in the pdf. In the right panel of that figure, we also show the marginal histograms of a parameter for every 200th iteration in the last 1000 iterations for a GSM run with batch size of 8, and these show that the distribution has indeed converged. A detailed discussion of this figure is presented in the general answers section above.
>
> We would also like to point out that these oscillations in KL divergence are also present in BBVI. Any stochastic optimization can dampen these by scheduling a learning rate to converge to 0, which is also what is often done in BBVI. While we have chosen not to do this for GSM, this same procedure can be applied to dampen the oscillations for GSM.

---

> > ### Comment · Reviewer_DLrt · 2023-08-17
> > **response rebuttal**
> >
> > Thank you for your detailed response regarding the concerns in my review. The added experiments and discussion sufficiently address the concerns and provides further empirical evidence for the soundness of the paper, and so I will raise my score to a 7.

---

### Official Review · Reviewer_h2vB · 2023-07-06

**Soundness:** 4 excellent
**Presentation:** 3 good
**Contribution:** 3 good
**Rating:** 7
**Confidence:** 4

**Summary:**

This paper proposes a novel alternative optimization strategy for approximate Bayesian inference in statistical modeling.
The starting point of the proposed score-based VI approach is the realization that two distributions are the same if their derivative is the same almost everywhere.
This principled is used to derive an objective to match the gradient of the log-density of the posterior and the gradient of the log-density of the approximate variational distribution.
The objective is set up in a way so as to promote a minimal change in the approximate variational distribution such that the scores are matched on a set of samples drawn from it.
Interestingly, in the Gaussian case this minimization has clased form.
This work is inspired by previous literature on Passive-Aggressive learning.


**Strengths:**

I believe that alternative optimization strategies for VI are an important area of research and, being very familiar with VI and unfamiliar with the literature on PA learning, I find the proposed idea quite original and well realized.
Someone may disagree that the idea is novel due to its derivation from PA learning, but in my opinion convincingly showing that this yields an effective strategy to obtain an approximation to the posterior distribution makes a good contribution to the literature.
The results are impressive, showing good performance compared to standard optimization (black-box VI) at a fraction of the number of gradient evaluations.


**Weaknesses:**

One weakness is that the work focuses heavily on the case when the approximation is Gaussian due to the close form solution of the proposed optimization strategy.
It would have been nice to get more insights into other approximations, e.g., Normalizing Flows.
Having said that, I believe that the Gaussian case is important and I understand that it was necessary to extensively study it in this version of the paper.


**Questions:**

The paper is very well written. However, I think that a bit more intuition on the algorithm with Eq 4 as objective would be ideal. At the moment, it seems a bit disconnected from Eq 3 - why would you want to minimally adjust q under that score-mathing constraint? A more comprehensive explanation could help here.
Also, even though it is written in some places, I think it is important to constantly remind the reader that there is no longer the traditional ELBO in the formulation, and that the optimization problem is really a different thing altogether.

I wonder about possible situations in which the optimization of Eq 4 could reach a bad local optimum. For example, it the initial q has variance which is too low, is it possible that samples have not enough diversity to allow for a proper coverage of the score-matching objective - and as a result this could result in the algorithm stopping with scores matched in a narrow region of the space with low posterior density.
I think that some insights into the behavior of the optimization wrt some of the choices that can be made in the initialization would add a lot to the revised version of the paper.

I was intrigued by the comments on the cases when the number of variational parameters is less than the number of parameters/latent variables. A bit more intuition on what is going on in these cases could also be useful for the revised version. And perhaps ways in which the proposed approach could be extended to handle these cases, which may not be that uncommon (e.g., a small hyper-net to generate large sets of parameters of a neural net).

I think that the experimental campaign is solid. However, it would have been really nice to try the proposed score-matching VI on largely parameterized models where the ELBO is known to be problematic. I wonder whether this could mitigate these effects due to the overparameterization (e.g., [1]).

[1] S. Rossi, P. Michiardi, and M. Filippone. Good Initializations of Variational Bayes for Deep Models. In Proceedings of the 36th International Conference on Machine Learning, ICML 2019, Long Beach, USA, 2019.


**Limitations:**

I haven't seen any specific text on the limitations of the approach and I think this could be included in the revision.

---

> ### Author Rebuttal · Authors · 2023-08-09
>
> > The paper is very well written. However, I think that a bit more intuition on the algorithm with Eq 4 as objective would be ideal. At the moment, it seems a bit disconnected from Eq 3 - why would you want to minimally adjust q under that score-mathing constraint?
>
> We can certainly add some more intuition. We know now from Lemma 1 that if Eq 3 holds for every $\theta$, then our variational family has found a perfect fit. Any practical method based on this observation can however only sample a batch of $\theta$’s at each iteration. When updating our variational parameters to satisfy the score matching equations over a given batch, we need to be careful that in the remainder of the domain (outside the batch), our variational family remains a reasonable fit. One way to ensure this is to always make the smallest possible change in our variational family to fit the score matching constraints over the current batch. This ensures that for the rest of the domain that has been explored before, the variational family will still be an approximately good fit. Thus the need to minimally adjust $q$
>  is because we only have partial information (a batch) of the constraints in (3). This approach is called the “least change” approach in quasi-Newton methods [Goldfarb] or the “passive-aggressive” approach in PA methods [Cramer].
>
> [Cramer] Online Passive-Aggressive Algorithms (JMLR 2006) \
> [Goldfarb]  Goldfarb, D. (1970), "A Family of Variable Metric Updates Derived by Variational Means", Mathematics of Computation, 24 (109): 23–26,
>
> > Though it is written in some places, I think it is important to constantly remind the reader that there is no longer the traditional ELBO in the formulation, and that the optimization problem is really a different thing altogether.
>
> We thank the referee for this suggestion and will put more emphasis on this in the revision
>
> > If the initial q has variance which is too low, is it possible that samples have not enough diversity to allow for a proper coverage of the score-matching objective - and as a result this could result in the algorithm stopping with scores matched in a narrow region of the space with low posterior density.  I think that some insights into the behavior of the optimization wrt some of the choices that can be made in the initialization would add a lot to the revised version of the paper.
>
> We agree with the referee’s intuition that mode collapse is a possibility in our algorithm. However this is not necessarily due to score matching, but rather due to generating samples from the variational distribution itself. This is also an issue with BBVI, but unlike BBVI, sampling in this way is not a requirement for GSM, but only convenience. If we have access to another distribution that covers the domain (for e.g. a prior distribution) and allows efficient sampling, we can always generate samples from it.
>
> This being said, we did investigate different initializations for both Gaussian and non-Gaussian targets. These included starting from–  a standard normal, random Gaussian with broad and narrow covariance, from the mode with identity or using an LBFGS approximation for the inverse covariance matrix, from the resulting approximation of the Pathfinder algorithm etc. \
> In all these cases, we always found the same, global solution for GSM. The only setting where the GSM solution depended on initialization was when the target distribution is mulit-modal with separated modes and in this case GSM generally converged to the nearest mode. However this is a known issue with almost all inference algorithms.  We can elaborate on this in the revised manuscript.
>
> > Comments on the cases when the number of variational parameters is less than the number of parameters/latent variables. A bit more intuition on what is going on in these cases could also be useful for the revised version. And perhaps ways in which the proposed approach could be extended to handle these cases, which may not be that uncommon (e.g., a small hyper-net to generate large sets of parameters of a neural net).
>
> Our setting is Gaussian families where there are $d^2+d$ variational parameters and $d$ latent variables. Thus the number of variational parameters is always greater than the number of  parameters/latent variables. In the general setting, for non-Gaussian variational families that have fewer parameters than latent variables, this score matching approach is not possible. But we are unaware of any practical examples where such a situation arises. Thus we now find that discussing this in our paper was a digression, and as such we will either remove this discussion, or expand upon it to make it clear. We also apologize that we are not familiar with this setting of “small hyper-net to generate large sets of parameters of a neural net”, and welcome a reference and are open to consider this during the discussion phase.
>
> > The experimental campaign is solid. However, it would have been really nice to try the proposed score-matching VI on largely parameterized models where the ELBO is known to be problematic. I wonder whether this could mitigate these effects due to the overparameterization.
>
> In this paper, we have focused only on the full-rank Gaussian variational family which cannot be extended to these highly over-parameterized models like deep networks. In the future, we plan on studying applications for mean-field Gaussian families (updates for which can be trivially implemented), and low-rank approximations. Both of these are better suited in the setting that the referee has pointed out. In that regard, we thank the referee for the reference and pointing to an interesting direction of research.
>
> > I haven't seen any specific text on the limitations of the approach and I think this could be included in the revision.
>
> We have discussed some limitations throughout the paper, especially in the conclusion and future work section, but will be happy to collect them in one sub-section in the revised paper.

---

> > ### Comment · Reviewer_h2vB · 2023-08-16
> >
> > Many thanks for your response - I don't think I will need any further clarifications.
> >
> > Having said that, after reading the other reviews, I wonder about the impact of the results with respect to BBVI with natural gradients. I'm looking forward to hearing the opinion of Reviewer CVEd to your response and to the discussion among reviewers.

---

> > > ### Author Response · Authors · 2023-08-17
> > >
> > > We thank the referee for their time and comments.
> > >
> > > > I wonder about the impact of the results with respect to BBVI with natural gradients. I'm looking forward to hearing the opinion of Reviewer CVEd to your response and to the discussion among reviewers.
> > >
> > > Based on the experiments with NGD-VI, both using our implementation with a Gauss-Newton approximation of the Hessian (VOGN updates) and the new package `gmmvi` that the reviewer CVEd pointed us to in the discussion, we find that GSM-VI consistently outperforms NGD-VI. Specifically, we find that while a properly tuned NGD-VI can be competitive with GSM in smaller dimension, it is very sensitive to parameter tuning and it does not scale well with respect to the dimension. This is on account of its Hessian approximation which becomes singular and unstable as the dimension grows. Furthermore, for both variants of NGD-VI, the method remains cubic in computational complexity, while GSM is quadratic.

---

### Official Review · Reviewer_Ciug · 2023-07-06

**Soundness:** 3 good
**Presentation:** 3 good
**Contribution:** 4 excellent
**Rating:** 7
**Confidence:** 4

**Summary:**

The paper proposes score matching as a new approach to (black box) variational inference (BBVI) where the variational family is Gaussian.
The usual way is to minimize the KL divergence (or equivalently to maximize the ELBO) using stochastic gradient descent (SGD) to update the variational parameters.
Instead, score matching imposes the constraint that the gradient of the log joint equal the gradient of the logarithm of the variational distribution for all parameter values.
The update step picks the new variational parameters to minimize the KL divergence between the new variational distribution and the previous variational distribution, under the score matching constraint at parameter values that are sampled from the previous variational distribution. The paper proves that this update step has a closed form solution if the variational family is Gaussian and can be computed in $O(d^2)$ time assuming the gradients can be computed in $O(d)$ time where $d$ is the dimension of the distribution.

The experimental evaluation compares the new Gaussian score matching VI (GSM-VI) with standard BBVI. If the target distribution is Gaussian, it finds that the number of iterations needed for convergence scales linearly with the dimension for GSM-VI, but worse for BBVI. It also investigates the effect of the condition number of the covariance matrix of the Gaussian target distribution and finds that it does not affect GSM-VI, but BBVI does not perform well if the condition number is large. Using a sinh-arcsinh normal distribution, the paper looks into what happens as the target distribution departs from being Gaussian. If the target is close to a Gaussian, then GSM-VI continues to require fewer gradient evaluations than BBVI, converging to a similar solution. If the target is far from being Gaussian, GSM-VI does not converge and can experience larger oscillations than BBVI. Finally, GSM-VI and BBVI are compared on real-world data from the posteriordb repository. In most of them, GSM-VI outperforms BBVI by a factor of 10 to 100 in terms of number of gradient evaluations.

**Strengths:**

The score matching idea is a novel approach to BBVI. It is a simple idea that is communicated clearly in the paper and shown to be very effective. It is clearly interesting and relevant to the variational inference community. The code for the experiments is available, which helps with the reproducibility of the results.

**Weaknesses:**

The main weakness of the paper is that it is almost purely empirical and provides no explanations for its (strong) empirical result: why is GSM-VI so effective? Why does it scale so well with the dimension? Why is a batch size of 2 best? (This seems particularly odd. I would expect batching to either provide no benefits or the benefit to increase with larger batch size.) Without even an attempt at an explanation, the empirical results almost seem too good to be true.

A potential weakness in the experimental evaluation is that the performance is always measured in terms of number of gradient evaluations, not actual running time. Given that the update step in GSM-VI takes $O(d^2)$ time, the paper should report the actual measured running times as well. It is also unclear if the quadratic update step will worsen GSM-VI's performance for very high-dimensional problems.

The experimental evaluation should include more details: how was GSM-VI implemented? Is it built on existing VI implementations? What kind of system was it run on? Not all (hyper-)parameters for the experiments were reported: what is the dimensionality of the real-world benchmarks? What was the batch size for BBVI? How were the posteriordb examples selected? And so on.

**Questions:**

These questions overlap with what I wrote under "Weaknesses". I copy them here.
- Why do you think GSM-VI is so effective and scales so well?
- What are the measured running times for the experiments (since the update step takes $O(d^2)$ time)?
- In a similar vein, does the $O(d^2)$ update step have a negative effect for very high-dimensional problems? Does BBVI overtake GSM-VI again for high dimensions in terms of running time?
- What was the batch size for BBVI in the benchmarks?
- What's the dimensionality of the posteriordb examples? How were they picked?
- Regarding figure 4: it would be interesting to see what happens for settings where both $s \ne 0$ and $t \ne 1$. Did you try this?

I will update my rating if the author's answers address my concerns.

**Limitations:**

The paper mentions some limitations in the text, but they are not collected in one place. It would be helpful if the paper elaborated on the limitations in a separate section/paragraph.

---

> ### Author Rebuttal · Authors · 2023-08-08
>
> > I will update my rating if the author's answers address my concerns.
>
> We thank the referee for their thoughtful reviews and kind consideration. Below, we address the weakness and questions raised above and will include this discussion in the revised manuscript.
>
> > Why do you think GSM-VI is so effective and scales so well?
>
> There are 3 intuitions that help explain why GSM-VI is more efficient than BBVI-
>
> 1) GSM-VI does not rely on Taylor approximations. BBVI is a stochastic gradient method and hence relies on the 1st order Taylor approximation of the objective function. In contrast, GSM-VI computes the exact projection onto the score matching constraint.
> 2) BBVI requires tuning a learning rate and its scheduling. In contrast, GSM-VI does not have a learning rate parameter. Instead, it is able to adaptively make large jumps in the initial iterations (see Fig. 1.a in the paper) and make smaller adjustments as the approximation converges to the target.
>  3) BBVI relies on a scalar signal, that is it attempts to increase the ELBO (scalar function) by using the steepest ascent direction (gradient). GSM-VI instead uses $d$ equations at each iteration to determine the update, one for each element of the score. Since the number of constraints increases linearly with dimensions, we believe it partly explains why GSM-VI scales well with dimension.
>
> > Why is a batch size of 2 best? This seems odd.
>
> Referee's intuition is correct. Our understanding of the impact of batch size has now improved with more experiments-
>
> - For Gaussian targets, GSM performs equally well for all $B\geq1$. There are minor gains for larger batches as the dimensionality increases $d\gtrapprox100$, but these are marginal enough that B=2 is a good conservative default.
> - For non-Gaussian targets, we now recommend a larger batch size. We find that these converge to a more stable solution i.e. smaller oscillations in KL divergence. We show this in Fig. 3 of the above attached pdf. We found that $B \geq 8$ suffices for $d\lessapprox100$
>
> > Performance is always measured in terms of number of gradient evaluations. The paper should also report actual running times.
>
> There are 2 reasons to measure gradient evaluations-
> 1. Actual running time is very sensitive to actual implementation details.
> 2. Often in real-world problems, the computational bottleneck is evaluating the target log density and its gradient (for instance it can involve solving complex ODEs). Hence we focus on minimizing these evaluations.
>
> However we acknowledge that the referee raises a good point. We have now timed GSM and BBVI updates for fitting a full rank Gaussian target with 2048 dimensions using batch size of 1 in Jax after JIT-compilation. GSM update takes 230 ms/iteration while BBVI takes 227 ms/iteration.
>
> > Will the quadratic update step will worsen GSM-VI's performance for high-dimensional problems. Does BBVI overtake GSM?
>
> In this paper, we only consider the Gaussian variational family with full-rank covariance matrix and hence both BBVI and GSM have the same $O(d^2)$ complexity. This complexity is unavoidable since we need to store and update the $d \times d$ covariance matrix.
> Thus based on our timing test for D=2048 problem above, and the fact that GSM requires $\sim10$x less gradient evaluations for almost all examples considered, we are confident that BBVI will not overtake GSM-VI as the dimension increases.
>
> > What was the batch size for BBVI in the benchmarks?
>
> We fixed the batch size of BBVI to 2 for all experiments, same as GSM, to keep the two algorithms at equal footing. Since we did not tune the batch size of GSM for individual experiments, we did not vary the batch size of BBVI either (however we did tune learning rate of BBVI).  In Figure 1 in the uploaded response pdf, we now show results for varying both- batch and learning rate of BBVI and find that GSM-VI still outperforms BBVI in all cases. Please see general answer for more details.
>
> > How was GSM-VI implemented? What kind of system was it run on?
>
> Our implementation of GSM-VI was in Tensorflow but now we have a Jax package that will be released after the review process. We will also implement GSM in BlackJax package. Also note that if one has access to the score function of the target, GSM updates can be written in native python without any ML or auto-diff. All the experiments were done on a single core of a personal CPU desktop.
>
> > How were the posteriordb examples selected?  What's their dimensionality?
>
> We compared GSM & BBVI on all the 50 models in PosteriorDB and found GSM to outperform on all of them. For brevity, we chose 8 of these models for the paper based on 2 considerations- i)  Each model represents a different class of statistical models, with different complexities, and ii) we have an equal representation of both Gaussian and non-Gaussian targets.
> All the models are low-dimensional which allows us to look at marginal and joint distributions of parameters and thus go beyond only comparing KL divergences in evaluating different algorithms.
> Similar choice of models was made by a recent paper on Pathfinder algorithm for fitting a Gaussian variational distribution (Lu et.al. JMLR ; 23(306):1−49, 2022)
>
> Classes and dimensionality of the 8 models used are- Generalized linear models (d=26), Differential equation dynamics (8), Hierarchical meta-analysis with centered & non-centered parameterization (10), hidden Markov models (8), time-series model (7),
> Gaussian processes (13), Gaussian mixture model (d=5).
>
> > Figure 4: what happens for settings where both $s\neq1$ and $t\neq1$.
>
> We did experiments varying both skewness and tail-weights at the same time for different dimensions and found that in all our experiments, GSM with batch $B=2$ always converged to a similar solution as BBVI, but faster. Hence for the sake of clarity, we separated these two axes and showed them separately. An example of $s\neq1$ & $t\neq1$ case is in Fig. 3 of the attached pdf.

---

> > ### Comment · Reviewer_Ciug · 2023-08-15
> >
> > Thank you very much for the thorough response. You have addressed essentially all my concerns, so I'm raising my rating from 5 to 7.

---

> > > ### Author Response · Authors · 2023-08-17
> > >
> > > We thank the reviewer for their time and comments, and kind consideration in raising the score.

---

### Author Rebuttal · Authors · 2023-08-08

We thank all the referee for their thoughtful reviews. Based on comments from different reviewers, we have now compiled three new figures from some old, and one entirely new experiment. Please see the attached pdf. These serve to answer some of the questions raised below and so we begin by discussing these figures here as a preamble. We will add these figures and discussion to the revised manuscript.

**Figure 1. Convergence of BBVI:**

In response to Reviewer **DLrt**'s question we have included an experiment tracking the reverse KL.
In Fig. 1 of the attached pdf, we fit a full-rank Gaussian target distribution of 16 dimensions with BBVI and GSM. Here we monitor the reverse (backward) KL divergence which is explicitly minimized by BBVI, and show results for doing a hyper-parameter search on both the parameters of BBVI- learning rate and the batch size. The top row shows reverse KL (y-axis) on linear scale and the bottom row on log-scale for clarity. We highlight two key takeaways from the figure:

1) Given enough computational budget, BBVI does converge to the same quality of solution as GSM, in the current example for larger batch sizes ($B\geq 4$) and smaller learning rates $lr \leq 0.01$ (see blue and orange lines in last two panels). However its performance is quite sensitive to the hyperparameters. The figure also suggests that performance of BBVI can be improved by scheduling the learning rate, but this will require a greater hyperparameter search.

2) Even for optimally tuned batch size and learning rate, GSM significantly outperforms BBVI and requires orders of magnitude less gradient evaluations for convergence even in terms of reverse KL, which is the metric that is explicitly minimized by BBVI.
We will include similar figures for other synthetic experiments in the revised version of the paper..

**Figure 2. Natural Gradient Descent (NGD)-VI**

We have added natural gradient descent VI for maximizing ELBO as a new baseline for comparison with GSM since it has been shown to significantly outperform BBVI. We indeed find this to be the case in our experiments. However as shown in Fig. 2, in our regimes of interest, we find that even compared to NGD-VI, GSM is faster in terms of gradient evaluations, has a smaller iteration complexity, and does not require tuning for variational inference with a full-rank Gaussian distribution.

*Implementation of NGD-VI:* We implemented the updates to variational parameters $\mu$ and $\Sigma$ based on Eq. 16 of Lin et.al (arxiv:1906.02914) (adapted for a single Gaussian instead of mixture model by setting $\delta_c=1$). These updates however require the Hessian of the objective function (ELBO). To approximate this with only first-order gradient information for comparison with GSM, we have used the VOGN update (Eq. 16 of Khan et.al., arxiv:1806.04854). This approximates the Hessian of the likelihood term $\log p(\theta | x)$ with a Gauss-Newton approximation and combines it with the correct Hessian of the entropy term $\log q(\theta)$- $\Sigma^{-1}$.

Figure 2 shows results of fitting a 64 dimensional Gaussian target with GSM and NGD-VI while varying its hyperparameters- the learning rate and the batch size. We monitor reverse KL, which is explicitly optimized by NGD-VI, for both linear (top) and log-scale (bottom row) on y-axis for clarity. We find that-
1) While an optimally tuned NGD-VI can be competitive with GSM, its performance is very sensitive to tuning the hyperparameters. This will require not only tuning the batch size and learning rate, but also the scheduling of learning rate. In comparison, GSM has no learning rate parameter to tune.
2) For large dimensions $d \geq 256$, Gauss-Newton approximation for Hessian in VOGN algorithm starts to diverge. This can possibly be corrected by using a regularization hyper-parameter for Hessian, or a different Hessian approximation, but both these require more tuning.
GSM does not face any such issues in scaling to high dimensions
3) Finally, we also note that for a  full-rank Gaussian variational family, every iteration of GSM has a quadratic complexity $d^2$ as compared to cubic $d^3$ of NGD-VI which requires one to estimate both $\Sigma$ and $\Sigma^{-1}$ for each update.

Based on these results, we conclude that  GSM-VI outperforms NGD-VI in our regimes of interest. We will add this baseline to other experiments in our revisions.

**Figure 3. Batch size, GSM-VI and non-Gaussian targets**

We have run more experiments with GSM-VI for different non-Gaussian targets and varied the batch size to better understand its convergence. Figure 3 summarizes our findings. In it, we compare the performance of GSM-VI on a synthetic non-Gaussian target (D=10) for different batch sizes (B=2, 8, 32) with BBVI (B=8, 32). We show the same forward KL divergence as the paper (left panel), marginal histogram for one of the parameters for all algorithms/configurations (middle panel) and additionally show the same marginal histogram for every 200th iteration in the last 1000 iterations of GSM-VI with B=8 to demonstrate that the distribution has indeed converged (right panel).

We find that larger batch sizes of GSM ($B \geq 8$)  lead to a more stable solution than both GSM with $B=2$ and BBVI, i.e. they have smaller (or no) oscillations in the forward KL metrics. GSM also converges to a point with a better forward KL divergence. This is also reflected in the marginal histograms which have larger dispersion for GSM than BBVI. Thus GSM does not lead to mode-seeking behavior. This can be desirable, for instance, as it means GSM posteriors can be corrected with importance sampling more easily.

---

### Decision · Program_Chairs · 2023-09-21

**Decision:**

Accept (poster)

**Comment:**

Thank you for your engaging interaction with the reviewers during the discussion. The reviewers and I are generally in agreement that this paper should be accepted. The paper is written very clearly, the methodology is novel and interesting (not just an iteration on existing variational methods), and the method seems to perform well on the baselines tested. However, there are no theoretical guarantees, and the range of baselines in the original submission was very limited.

In the camera ready, the paper should be clear about the lack of analysis, and provide a more extensive set of comparisons to existing approaches with a fair and unbiased analysis of results (among other edits suggested by reviewers) to be suitable for publication. The authors have already done some work in this direction during the discussion phase, so I am confident in the authors' ability to complete this work for the camera ready.